# Dual leucine zipper kinase-dependent PERK activation contributes to neuronal degeneration following insult

Martin Larhammar[1], Sarah Huntwork-Rodriguez[1†], Zhiyu Jiang[1], Hilda Solanoy[1†], Arundhati Sengupta Ghosh[1], Bei Wang[1‡], Joshua S Kaminker[2], Kevin Huang[2], Jeffrey Eastham-Anderson[3], Michael Siu[4], Zora Modrusan[5], Madeline M Farley[6], Marc Tessier-Lavigne[1,7§], Joseph W Lewcock[1*†], Trent A Watkins[1,6,8*]

[1]Department of Neuroscience, Genentech, Inc., San Francisco, United States; [2]Bioinformatics, Genentech, Inc., San Francisco, United States; [3]Pathology, Genentech, Inc., San Francisco, United States; [4]Discovery Chemistry, Genentech, Inc., San Francisco, United States; [5]Molecular Biology, Genentech, Inc., San Francisco, United States; [6]Department of Neurosurgery, Baylor College of Medicine, Houston, Texas; [7]Laboratory of Brain Development and Repair, The Rockefeller University, New York, United States; [8]OMNI Biomarkers Development, Genentech, Inc., San Francisco, United States

*For correspondence: lewcock@ dnli.com (JWL); trentw@bcm.edu (TAW)

Present address: †Denali Therapeutics, South San Francisco, United States; ‡Weill Institute for Cell and Molecular Biology, Cornell University, Ithaca, United States; §Office of the President, Stanford University, Stanford, United States

**Abstract** The PKR-like endoplasmic reticulum kinase (PERK) arm of the Integrated Stress Response (ISR) is implicated in neurodegenerative disease, although the regulators and consequences of PERK activation following neuronal injury are poorly understood. Here we show that PERK signaling is a component of the mouse MAP kinase neuronal stress response controlled by the Dual Leucine Zipper Kinase (DLK) and contributes to DLK-mediated neurodegeneration. We find that DLK-activating insults ranging from nerve injury to neurotrophin deprivation result in both c-Jun N-terminal Kinase (JNK) signaling and the PERK- and ISR-dependent upregulation of the Activating Transcription Factor 4 (ATF4). Disruption of PERK signaling delays neurodegeneration without reducing JNK signaling. Furthermore, DLK is both sufficient for PERK activation and necessary for engaging the ISR subsequent to JNK-mediated retrograde injury signaling. These findings identify DLK as a central regulator of not only JNK but also PERK stress signaling in neurons, with both pathways contributing to neurodegeneration.

## Introduction

Neuronal stress response pathways are engaged following acute neuronal injury and in chronic neurodegenerative disease. The resulting regulation of gene expression controls both cellular repair processes and apoptosis (*Scheper and Hoozemans, 2013*; *Tedeschi and Bradke, 2013*). Understanding the mechanisms underlying these stress responses may therefore offer opportunities for therapeutic intervention. One such pathway is the Integrated Stress Response (ISR), which is thought to be engaged in models of neurodegenerative diseases by PERK, one of four eukaryotic initiation factor 2 alpha (eIF2α) kinases (*Halliday and Mallucci, 2014*). The resulting phosphorylation of eIF2α suppresses general translation but enhances the translation of select mRNAs, including that of the *Activating transcription factor-4* (*Atf4*), *Protein phosphatase 1 regulatory subunit 15A* (*Ppp1r15a*, which encodes the protein Gadd34), and *DNA damage-inducible transcript-3* (*Ddit3*, which encodes the protein CHOP) (*Palam et al., 2011*; *Wek et al., 2006*). Disrupting PERK signaling is

neuroprotective in models of Alzheimer's disease (AD), prion diseases, and other disorders involving protein aggregation (*Halliday et al., 2015*; *Ma et al., 2013*; *Matus et al., 2013*; *Moreno et al., 2013*). Canonically, PERK is stimulated by endoplasmic reticulum (ER) stress and initiates the ISR as one of the three branches, along with IRE1α and ATF6α, of the Unfolded Protein Response (UPR) (*Wek et al., 2006*), although the upstream regulators of PERK activation in the context of disease have not yet been determined.

Whereas emerging evidence indicates that neuronal PERK activity may be a feature of progressive neurodegenerative conditions, the role of PERK following acute neuronal insults remains poorly understood. Components and regulators of the UPR, including phosphorylated PERK, phosphorylated IRE1α, CHOP, and the ATF6-related transcription factor Luman, have recently been noted after nerve damage, with Luman serving a pro-regenerative role (*Hu et al., 2012*; *Yang et al., 2016*; *Ying et al., 2015*). However, a seemingly distinct cellular stress response pathway driven by the mixed lineage kinase DLK appears to be a predominant factor in determining neuronal fate in these settings. Following neurotrophin deprivation or axonal damage, DLK stabilization leads to the activation of the stress-responsive c-Jun N-terminal kinases JNK2 and JNK3 (*Sengupta Ghosh et al., 2011*; *Huntwork-Rodriguez et al., 2013*; *Tedeschi and Bradke, 2013*). Downstream of JNK2 and JNK3 kinase activity, phosphorylation of the AP-1 transcription factor c-Jun leads to the induction of multiple regeneration-associated genes, and this pathway is required for robust axon regrowth after peripheral nerve injury (*Fernandes et al., 2013*; *Raivich et al., 2004*; *Shin et al., 2012*). Interestingly, c-Jun also regulates the survival of injured neurons, perhaps in part through transcription of ISR-associated genes. For instance, it is necessary for the upregulation of the UPR-associated mRNA *Gadd45a* that is important for neuronal survival after peripheral nerve damage (*Fernandes et al., 2013*; *Lin et al., 2011*). c-Jun also contributes to the expression of a number of pro-apoptotic genes, including the UPR-associated mRNAs *Ddit3* and *Atf3* that contribute to the death of axotomized retinal ganglion cells (RGCs) following optic nerve crush (ONC) injury (*Fernandes et al., 2013*; *Hu et al., 2012*). JNK2/JNK3 or c-Jun deficiency is neuroprotective over the first two weeks after ONC but does not enable the sustained RGC survival provided by DLK deficiency (*Fernandes et al., 2012*; *Watkins et al., 2013*; *Welsbie et al., 2013*), raising the possibility that DLK controls additional signaling pathways. In the current study, our efforts to understand neuronal stress signaling pathways after acute injury uncovered a functional role for PERK signaling in regulating apoptotic responses downstream of DLK.

## Results

### ISR-related expression changes in both PNS and CNS models of axonal damage

To explore pathways induced by acute neuronal stress, we began by evaluating a model of peripheral nerve injury that is known to initiate transcriptional stress responses in the axotomized neurons residing within the dorsal root ganglion (DRG) (*Figure 1a*). Expression profiling of lumbar (L4) DRG 24 hr after sciatic nerve transection revealed a number of injury-regulated mRNAs resembling those observed in similar studies of both peripheral and CNS axonal damage (*Hu et al., 2016*; *Li et al., 2015*; *Stam et al., 2007*; *Yang et al., 2007*; *Yasuda et al., 2014*), including our previous study of ONC (*Watkins et al., 2013*) (*Figure 1b*). In addition to known c-Jun-dependent expression changes (*Fernandes et al., 2013*), in both models we noted the presence of ISR-associated mRNAs, including *Atf4* and its target genes *Chac1* and *Eif4ebp1* (*Figure 1b* and *Figure 1—figure supplement 1*) (*Mungrue et al., 2009*; *Yamaguchi et al., 2008*), as well as *Eif2ak3* (*Figure 1—figure supplement 2*), but not other eIF2α kinases (*data not shown*). Further analysis comparing nerve injury-induced mRNAs to genes known to be upregulated in an ATF4-dependent manner after neuronal oxidative stress (*Lange et al., 2008*) revealed that the expression changes observed in this ISR-related gene set is greater than would be expected by chance in both axonal injury models (*Figure 1c,d*). Together these observations suggested the hypothesis that the acute neuronal stress response may include activation of PERK signaling in multiple settings, consistent with a report that nerve crush leads to PERK phosphorylation (*Ying et al., 2015*).



**Figure 1.** Acute neuronal insults activate the integrated stress response (ISR). (**a**) Schematic of sciatic nerve crush (SNC) and optic nerve crush (ONC). Mouse lumbar level 3 and 4 dorsal root ganglia (DRG) and retina were isolated to assess the neuronal stress response after SNC and ONC, respectively. (**b**) Microarray cross-comparison of injury-regulated mRNAs following SNC or ONC (n = 5 per condition) identifies multiple ISR-associated genes (blue), including *Eif4ebp1*, *Atf4*, *Chac1* and *Eif2ak3* (PERK), upregulated by both insults. (**c–d**) mRNAs within the 'ISR-related' gene set (see Materials and Methods) are observed more frequently amongst upregulated mRNAs than expected by the overall distribution of mRNA expression changes assessed in each microarray study ('complete gene set') following SNC (p=2.4 × $10^{-5}$, (**c**)) or ONC (p=9.9 × $10^{-7}$, (**d**)), suggesting the selective activation of the ISR. (**e–f**) Immunoblots reveal upregulation of the ISR (p-PERK, p-eIF2α, and ATF4), in addition to the JNK-pathway (p-c-Jun) in L3/L4 DRG lysates after SNC (**e**), and in retina lysates after ONC (**f**). The time post-injury is indicated in hours. (**g**) Primary e12.5 mouse DRG cultures deprived of NGF (3 h) or treated with the ER stress inducer thapsigargin (Tgn) in the vehicle DMSO. NGF deprivation engages phosphorylation of PERK (p-PERK/PERK), p-eIF2α and ATF4. Protein levels were normalized to GAPDH and non-NGF deprived vehicle control (n = 6–7/condition, four independent experiments). (**h**) siRNA targeting each of the four eIF2α kinases differentially impacts ISR activation at 3 h after NGF withdrawal from embryonic DRG neuronal cultures, with only siRNA targeting *Eif2ak3* consistently reducing ATF4 protein levels. (**i**) Representative TUJ-1 immunostainings 42 hr after isolation and siRNA-transfection of adult DRG neurons. (**j**) OnTarget Plus siRNA-mediated knockdown of *Atf4* or *Eik2ak3*, but not other eIF2α kinases, enhances adult sensory axon regrowth in vitro (n ≥ 8 wells/condition). Molecular weight indicated in kilodaltons (kDa). Data are represented as mean ± SEM. *p<0.05, **p<0.01, ***p<0.001, ****p<0.0001, one-way ANOVA with post-hoc Bonferroni test.

The following figure supplements are available for figure 1:

*Figure 1 continued on next page*

*Figure 1 continued*

**Figure supplement 1.** Heat maps showing expression of 30 ISR-related mRNAs that are induced by neuronal oxidative stress in an ATF4-dependent manner (*Lange et al., 2008*) in L4 DRG after sciatic nerve transection (a) or retina after optic nerve crush (b, data from *Watkins et al., 2013*) (n = 4 per condition).

**Figure supplement 2.** qRT-PCR validation of *Perk* mRNA upregulation in L4 DRG 24 h after SNC using two different primer sets (n = 4 per condition).

**Figure supplement 3.** Targeted silencing of ATF4 by OnTarget Plus siRNA pool verified by immunoblotting.

**Figure supplement 4.** Immunohistochemistry of DRG cryosections from *Syn1-Cre;R26^LSL.tdTomato* mice exhibits labeling of axons from medium- and large-diameter TUJ1-positive sensory neurons.

**Figure supplement 5.** Assessment of PERK influence on axon regeneration in vivo after SNC.

## Acute neuronal insults activate PERK

To further investigate whether PERK signaling is engaged by axonal injury, we first evaluated markers of pathway activation in protein lysates of L3/L4 DRGs after sciatic nerve crush (SNC). In addition to the expected phosphorylation of c-Jun indicative of stress-responsive JNK signaling, immunoblot analysis revealed a robust elevation of the transcription factor ATF4 after injury (*Figure 1e*), consistent with activation of the ISR. We also noted potential increases in the levels of PERK, the phosphorylated form of PERK (p-PERK) and its target eIF2α (p-eIF2α) over multiple experiments and time points, although as noted previously in assessments of PERK signaling (*Qi et al., 2011*), the changes in these markers were less definitive than those observed with ATF4. We found a similarly significant upregulation of ATF4 in retina following ONC, along with more modest and variable elevation of p-PERK and p-eIF2α (*Figure 1f*). Together, the induction of ATF4 protein following both SNC and ONC provided further support for ISR activation as a feature of the acute axonal injury response. However, with only a small fraction of cells in each tissue representing axotomized neurons, we found quantitative assessment of other markers of PERK signaling to be of limited utility.

Given these limitations and the possibility that PERK activation is a general characteristic of acute neuronal stress, we hypothesized that quantitative analysis of the PERK pathway may be more suitable utilizing a distinct in vitro model. Withdrawal of nerve growth factor (NGF) from cultured embryonic DRG neurons initiates an acute stress response that shares several aspects with adult axonal injury responses (*Huntwork-Rodriguez et al., 2013*; *Yang et al., 2013*). We observed that NGF withdrawal results in induction of not only JNK signaling, as previously described (*Sengupta Ghosh et al., 2011*), but also the ISR, as shown by activation of PERK and ATF4 (*Figure 1g*), and an elevation in the phosphorylated form of eIF2α. Though not as substantial as PERK activation in response to thapsigargin (Tgn), a positive control that potently induces ER stress (*Booth and Koch, 1989*), these responses provided additional evidence that acute neuronal insults engage PERK signaling.

We next determined whether ATF4 upregulation following NGF withdrawal is dependent on PERK activity or is controlled by other ISR kinases, a question that we addressed using two distinct in vitro stress response assays, one employing embryonic neurons and one employing adult neurons. In the first, utilizing the embryonic DRG culture model described above, we targeted each of the four eIF2α kinases with distinct siRNA pools following NGF deprivation. Whereas targeting *Eif2ak1*, *Eif2ak2*, or *Eif2ak4* resulted in modest, variable effects on ATF4 protein levels, knockdown of *Eif2ak3* consistently abrogated the upregulation of ATF4, suggesting that PERK is largely responsible for this effect (*Figure 1h*). In the second approach, we probed the role of ISR kinases in adult axonal injury. Isolation of adult DRG neurons axotomizes them and serves as an in vitro model of peripheral nerve stress signaling and axon regeneration (*Smith and Skene, 1997*). By quantifying neurite growth of these adult sensory neurons on a growth-restrictive substrate over two days in culture, we found that knockdown of ATF4 enhances axon regeneration (*Figure 1i* and *Figure 1—figure supplement 3*). Although we and others have found that disruption of the ISR is not sufficient to impact regeneration in vivo (*Oñate et al., 2016*) (*Figure 1—figure supplement 4*, *Figure 1—figure supplement 5*, and *data not shown*), this system provided a sensitive bioassay that revealed that siRNAs

targeting *Eif2ak3* but not other eIF2α kinases mimicked the effect of siRNA targeting *Atf4*, with no additive effect of targeting both (*Figure 1j*). These findings in two different models do not rule out contributions from other eIF2α kinases but focused our attention on PERK as the primary regulator of ATF4 in acute neuronal insults.

## The ISR influences the mRNA levels of ATF4 target genes after nerve injury

We next assessed how ISR-mediated upregulation of ATF4 contributes to stress-induced mRNA changes and whether this pathway functions similarly in both an adult injury model (SNC) and a developmental neurodegeneration model (NGF deprivation). First, we investigated the ATF4-dependence of a selection of four transcripts previously found to be associated with the ISR that were consistently upregulated in our microarray studies (*Figure 1b*) (*Mungrue et al., 2009*; *Wek et al., 2006*; *Yamaguchi et al., 2008*). qRT-PCR analysis of L4 DRG showed the persistent upregulation of eIF4E binding protein-1 (*Eif4ebp1*) and *Chac1* mRNA over the first three days, whereas *Ddit3* and *Ppp1r15a* mRNA, initially elevated, returned to near baseline levels by the second day (*Figure 2a*). Evaluation of L4 DRGs from ATF4-deficient (*Yoshizawa et al., 2009*) and wild-type control mice (see Supplemental Methods, *Figure 2—figure supplement 1*, and *Figure 2—figure supplement 2*) revealed at least partial ATF4-dependence of *Eif4ebp1* and *Chac1* but not *Ddit3* and *Ppp1r15a* (*Figure 2b*). Although some typical ATF4 transcriptional targets (*Ddit3*, *Ppp1r15a*) are upregulated independently of ATF4 in SNC, these findings suggest that ATF4 contributes to the regulation of gene expression after acute axonal injury.

Based on our evidence of PERK signaling, we hypothesized that the upregulation of functional ATF4 protein is not simply a consequence of induction of *Atf4* mRNA (*Figure 1b*) but, perhaps more significantly, a function of enhanced translation of *Atf4* mRNA upon activation of the ISR. To evaluate this hypothesis, we utilized a recently developed inhibitor of this pathway, the I̲ntegrated S̲tress R̲esponse I̲nhibitor (ISRIB), a compound that blocks stress-induced translation of *Atf4* mRNA (*Sidrauski et al., 2013*, *2015*). ISRIB administration prevented the upregulation of ATF4 protein in L3/L4 DRG after SNC but did not alter p-c-Jun induction (*Figure 2c*). Consistent with this observation and our data in ATF4-deficient mice, ISRIB treatment significantly reduced *Chac1* and *Eif4ebp1* mRNA, but not *Ppp1r15a* and *Ddit3* mRNA in L4 DRG following SNC (*Figure 2d*). Together, these data implicate ISR-mediated regulation of ATF4 in the gene expression changes induced by axonal damage.

To determine whether the ISR similarly regulates expression changes following NGF deprivation, we evaluated ISRIB-treated embryonic DRG cultures for the same putative ATF4 target genes. As was observed following SNC, all four mRNAs are induced by NGF withdrawal, yet the regulation of these targets in this model exhibits distinct ISR-dependence. In this instance, not only *Chac1* and *Eif4ebp1* mRNA, but also *Ppp1r15a* and *Ddit3* mRNA, exhibit ISRIB sensitivity (*Figure 2e*). Together, these data suggest both commonalties and differences in the role of the ISR in gene expression across distinct types of neuronal stress.

## DLK/JNK signaling is required for PERK activation

To better understand the context of PERK signaling in acutely damaged neurons, we explored potential relationships between the ISR and the seemingly distinct DLK/JNK/c-Jun cascade that potently regulates neuronal degeneration in the NGF withdrawal (*Sengupta Ghosh et al., 2011*) and nerve injury models (*Watkins et al., 2013*; *Welsbie et al., 2013*). In considering whether PERK may mediate neurodegeneration through stimulation of DLK/JNK signaling, we noted that ISRIB treatment did not impact the levels of phospho-c-Jun following SNC, suggesting that DLK/JNK signaling is not downstream of PERK (*Figure 2c*). Accordingly, the induction of p-c-Jun is unaltered in ATF4-deficient mice (*Figure 2—figure supplement 2*). This suggests that disruption of the ISR is insufficient to block DLK/JNK-mediated signaling.

We next assessed whether, conversely, DLK disruption influences PERK signaling. For this purpose, we first utilized the SNC model in tamoxifen-inducible DLK-deficient mice (*CAG^ERT-Cre*; *Map3k12^lx/lx*) (*Watkins et al., 2013*). Immunoblots of L3/L4 DRG demonstrated that the absence of DLK prevents not only c-Jun phosphorylation but also upregulation of ATF4 after SNC (*Figure 3a* and *Figure 3—figure supplement 1*). Consistent with this, qRT-PCR analysis revealed that not only



**Figure 2.** The PERK-mediated ISR regulates mRNA levels of a subset of ATF4 target genes. (**a**) qRT-PCR of L4 DRG confirms the upregulation of each of four ISR-related mRNAs (*Chac1*, *Eif4ebp1*, *Ppp1r15a*, and *Ddit3*) over the first 3 days after SNC (n = 3/condition). (**b**) Real-time qPCR for four putative ATF4 target genes in ATF4-null and wild-type mouse DRGs 24 h following SNC (n = 3/condition). (**c–d**) Dosing with ISRIB reduces induction of ATF4 protein and the mRNA of a subset of its putative target genes in associated DRGs following SNC. (**c**) Immunoblots of L3/L4 DRGs from ISRIB-treated mice 16 h post-SNC (n = 3/condition). (**d**) Real-time qPCR of mouse DRGs 16 h following SNC (n = 3/condition). Mice were dosed with vehicle or ISRIB (10 mg/kg, 1 h pre-SNC and 12 h post-crush). (**e**) qPCR of NGF-deprived embryonic DRG cultures reveals upregulation of mRNA of four putative ATF4 target genes, blocked by 400 nM ISRIB (n = 6–8/condition, two independent experiments). Data are represented as mean ± SEM. One-way ANOVA with post-hoc Bonferroni was used for statistical comparisons. *p<0.05, **p<0.01, ***p<0.001.

The following figure supplements are available for figure 2:

**Figure supplement 1.** The structure of a targeted ATF4 allele (*Yoshizawa et al., 2009*), indicating the placement of a loxp-flanked *neo* cassette at the 3' end of the exon 1 (E1).

**Figure supplement 2.** Immunoblots indicate a lack of ATF4 protein induction in L3/L4 DRG 24 hr post-SNC in ATF4[loxp/loxp] mice, irrespective of the presence of *Cre* recombinase.

ATF4-independent (*Ppp1r15a* and *Ddit3*) but also ATF4-mediated expression changes (*Eif4ebp1* and *Chac1*) exhibited DLK-dependence (*Figure 3b*). These results indicate that DLK is necessary for ISR-mediated effects following adult nerve injury in vivo.

To determine whether these observations generalize to other types of acute neuronal stress, we returned to the NGF deprivation model. Immunoblots of embryonic DRG neuron lysates revealed that DLK kinase inhibition using the selective inhibitor GNE-3511 (*Patel et al., 2015*) resulted in not only reduced p-c-Jun but also suppressed PERK phosphorylation and ATF4 upregulation (*Figure 3c*



**Figure 3.** DLK is necessary and sufficient for PERK activation. (a–b) DLK-deficient (*CAG-ERT^pos^;Map3k12^lx/lx^*, DLK cKO) mice exhibit reduced activation of the ISR and JNK pathways following SNC compared to controls (*CAG-ERT^neg^;Map3k12^lx/lx^*). (a) Activation of ATF4 is reduced in DLK cKO L3/L4 DRGs 24 h after SNC (n = 3/condition). (b) qRT-PCR of ISR-associated genes in L4 DRG following SNC in DLK cKO mice compared to controls (n = 4/condition). (c) Inhibition of DLK (DLKi, 500 nM) reduces p-PERK and ATF4 induction, in addition to p-c-Jun, compared to DMSO vehicle controls following NGF-withdrawal in primary DRG cultures. ISRIB (500 nM) treatment specifically reduces ATF4 upregulation. (n = 6/condition, aggregated data from four independent experiments). Thapsigargin (Tgn, 100 nM) provides a positive control for PERK activation. (d) Primary DRG cultures deficient in DLK (DLK KO) lack activation of PERK and ATF4 following NGF deprivation (n = 4/condition). (e) Doxycycline (DOX)-inducible DLK expression in stably-transfected HEK293 cells activates PERK (assessed by molecular weight shift, as anti-p-PERK does not recognize human p-PERK) and ATF4, which can be blocked by DLKi or ISRIB treatment. (f) Transient transfection of mouse wildtype DLK, but not kinase dead DLK-S302A control, in HEK293T cells activates the ISR. One-way ANOVA with post-hoc Bonferroni was used for statistical comparisons. *p<0.05, **p<0.01, ***p<0.001, ****p<0.0001.

The following figure supplements are available for figure 3:

**Figure supplement 1.** Quantification of p-c-Jun immunoblots presented in *Figure 3a* (DLK-deficient L3/L4 DRGs 24 hr after SNC, n = 3/condition).

**Figure supplement 2.** Quantification of p-c-Jun immunoblots presented in *Figure 3c* (NGF-withdrawal in primary DRG cultures, n = 6/condition, aggregated data from four independent experiments).

**Figure supplement 3.** Quantification of p-c-Jun immunoblots presented in *Figure 3d* (NGF-withdrawal in primary DRG cultures, n = 4/condition).

and *Figure 3—figure supplement 2*). In contrast, NGF-deprived cultures treated with ISRIB specifically displayed reduced ATF4, but not p-PERK and p-c-Jun, as would be predicted for inhibition of the ISR downstream of both DLK and PERK (*Figure 3c* and *Figure 3—figure supplement 2*). DLK-deficient embryonic DRG neurons (*Sengupta Ghosh et al., 2011*) confirmed the results observed with DLK inhibitor and displayed no induction of p-PERK or ATF4 following NGF withdrawal (*Figure 3d* and *Figure 3—figure supplement 3*). Together, these data argue that DLK activation is necessary for PERK signaling following NGF withdrawal.

To determine if DLK is also sufficient to engage PERK signaling, we examined the effects of ectopic DLK expression in non-neuronal cells that lack endogenous DLK. We found that DLK overexpression in a doxycycline-inducible stable 293T cell line (*Huntwork-Rodriguez et al., 2013*) results in both elevated ATF4 protein and a molecular weight shift in human PERK indicative of phosphorylation (*Axten et al., 2012*) (*Figure 3e*). DLK activity is required for this effect, as increasing concentrations of DLK inhibitor reduced DLK-mediated activation of PERK, c-Jun, and ATF4, whereas ISRIB selectively blocked ATF4 induction (*Figure 3e*). Furthermore, HEK293T cells expressing wildtype DLK, but not a kinase-dead mutant (S302A) of DLK, resulted in upregulation of ATF4 protein levels (*Figure 3f*). Taken together, these findings demonstrate that DLK activity is both necessary and sufficient for initiating both the ISR and JNK signaling, with both contributing to neurodegeneration following acute insults. Activation of the ISR is therefore a novel component of the broad DLK-mediated stress response.

## Axonal JNK signaling contributes to PERK activation upon neuronal stress

We next investigated where the ISR is engaged along the DLK/JNK/c-Jun cascade that transmits local axonal injury signals to the nucleus (*Sengupta Ghosh et al., 2011*). First, we assessed whether activation of the ISR requires JNK signaling by examination of knockout mice lacking both of the stress-responsive JNK isoforms JNK2 and JNK3 that act downstream of DLK following injury (*Bogoyevitch, 2006*; *Sengupta Ghosh et al., 2011*). As observed for DLK-deficiency, we found suppression of ATF4 protein upregulation in JNK2/3-deficient DRGs after SNC (*Figure 4a* and *Figure 4—figure supplement 1*). These results demonstrate that both DLK and JNK are necessary for ISR activation after injury.

DLK/JNK-mediated retrograde injury signaling is important in both in vitro and in vivo models of axonal insults (*Cavalli et al., 2005*; *Sengupta Ghosh et al., 2011*), and retrograde signaling of axonally-translated ATF4 has been implicated in models of Alzheimer's disease (*Baleriola et al., 2014*). Accordingly, we next investigated whether PERK signaling following acute insults originates in axons at site of injury or subsequent to retrograde signaling. First, we attempted to assess whether elevated p-PERK and ATF4 protein could be observed in sciatic nerve segments proximal to the injury site after SNC (*Figure 4b*). In contrast to the elevated p-PERK and ATF4 levels observed in L3/L4 DRGs, associated sciatic nerve segments did not exhibit clear evidence of ATF4 and p-PERK, and the marginal signal sometimes detected in nerve could potentially represent induction of the ISR in neighboring cells rather than axons (*Mantuano et al., 2011*) (*Figure 4b*). To examine axonal activation independently of surrounding cells, we next established embryonic DRG explant cultures on membranes in which the axons grow through the pores of the membrane allowing for isolation of axons from cell bodies (*Unsain et al., 2014*). Although p-PERK and ATF4 increase over the first six hours following NGF withdrawal in the chamber containing cell bodies and proximal axons, the corresponding distal axons did not exhibit detectable levels of either (*Figure 4c*).

To evaluate the possibility that sub-detectable levels of ISR activation at the site of axonal insult are important, we next established compartmentalized Campenot chambers (*Campenot, 1977*), in which distal axons can be locally deprived of NGF (*Figure 4d*). Consistent with our observations following nerve crush in vivo, we observed that distal axonal NGF deprivation is sufficient to result in elevated ATF4 protein in the central compartment containing NGF-treated DRG cell bodies and proximal axons (*Figure 4d*). Inhibition of JNK signaling in either the central or distal chambers prevented the induction of ATF4. In contrast, ISRIB only had this effect when applied centrally (*Figure 4d*). Together, these data do not preclude a role for low levels of axonally translated ATF4 in regulating acute injury responses (*Baleriola et al., 2014*) but suggest a prominent contribution of PERK activity that occurs after JNK-mediated retrograde signaling.



**Figure 4.** PERK activation is dependent on JNK-mediated retrograde signaling but not c-Jun. (**a**) Immunoblot of JNK2/3-deficient (*Mapk9*$^{-/-}$ *Mapk10*$^{-/-}$, *Mapk9/10* KO) L3/L4 DRGs 24 h after SNC demonstrates reduced ATF4 induction compared to double heterozygous controls (*Mapk9*$^{+/-}$ *Mapk10*$^{+/-}$, *Mapk9/10* HET) (n = 3/condition). (**b**) Immunoblots of L3/L4 DRGs and 5-mm sciatic nerve (SN) segments 2–3 mm proximal to the injury site at 20 h post-SNC (+) illustrate elevation of DLK within the nerve and associated DRGs compared to uninjured DRGs and SN. PERK signaling is not detected within the SN after injury, in contrast to the associated L3/L4 DRGs. (**c**) Illustration of experimental setup for the isolation of DRG axons from explants (left, see methods). Immunoblots of DRG explants and enriched axons following NGF deprivation at the indicated time points. Phosphorylation of PERK and induction of ATF4 and p-c-Jun is observed in explant (grey), but not axonal (white), lysates quantified by immunoblotting (n = 5/condition, two experiments). (**d**) Schematic of compartmentalized Campenot chamber setup (top, see methods) displaying isolated axons in the outer compartment and cell bodies/proximal axons in the central compartment. Immunoblots of central compartment lysates after axonal NGF deprivation display elevated ATF4 levels. JNKi and DLKi prevent ATF4 induction when applied to either the central somal compartment or outer axonal compartments, whereas ISRIB only reduces ATF4 upregulation when applied to the central compartment (n = 6–9/condition, five independent experiments). (**e**) Immunoblots of siRNA-transfected embryonic DRG neurons suggests that c-Jun does not contribute to PERK activation following NGF withdrawal (n = 4–5/condition, three experiments). One-way ANOVA with Bonferroni's post-hoc test was used for statistical analysis. Data are mean ± SEM. *p<0.05, **p<0.01, ****p<0.0001.

The following figure supplements are available for figure 4:

**Figure supplement 1.** Quantification of p-c-Jun immunoblots presented in *Figure 4a* (JNK2/3-deficient L3/L4 DRGs 24 hr after SNC compared to HET controls, n = 3/condition).

**Figure supplement 2.** Quantification of p-c-Jun immunoblots presented in *Figure 4e* (NGF-withdrawal in siRNA-transfected primary DRG cultures, n = 4–5/condition, three experiments).

The observation that ISR activation occurs subsequent to retrograde signaling raised the possibility that injury-induced transcriptional changes downstream of c-Jun may mediate PERK activation. We therefore asked if upregulation of ATF4 protein is influenced by the levels of c-Jun, finding that reducing c-Jun expression by siRNA in DRG neurons does not reduce induction of ATF4 upon NGF deprivation (*Figure 4e* and *Figure 4—figure supplement 2*). Although a small increase in ATF4 transcription can be observed following nerve crush (*Figure 1b*), these results are consistent with the central role for translational control of ATF4 implied by ISRIB sensitivity and suggest that DLK does not control activation of the ISR primarily through c-Jun-mediated transcriptional changes.

## Profiling of ISR-dependent gene expression changes following NGF deprivation

The preceding data argues that ISR activation by PERK represents a previously unappreciated general feature of the DLK-mediated stress response that is more commonly associated with MAP kinase stress signaling. To more broadly examine the relative contribution of DLK/ISR signaling, we performed high-throughput RNA sequencing (RNA-seq) on NGF-deprived cultures of embryonic DRG sensory neurons in the presence of either ISRIB or DLKi (*Figure 5*).

First, we assessed whether global expression analysis supports the hypothesis that SNC, ONC, and NGF deprivation share the ISR as a common feature. Cross-model analysis showed less broad commonality in expression patterns when comparing SNC to NGF deprivation (*Figure 5—figure supplement 1*) or when comparing ONC to NGF deprivation (*Figure 5—figure supplement 2*) than we observed when comparing SNC to ONC (*Figure 1b*). Nevertheless, we found that, as in the ONC and SNC models, an ISR-related gene set representing putative ATF4-dependent transcripts (*Lange et al., 2008*) is enriched in this neuronal stress model (*Figure 5a*). We further examined the ISR genes that we had previously evaluated by qPCR, confirming that *Chac1*, *Eif4ebp1*, *Ppp1r15a*, *Ddit3* were, along with *Jun*, among a group of 1130 mRNAs that reached a strict criterion (>1.5 fold, adjusted p<0.001) for expression change following NGF deprivation (*Figure 5b* and *Figure 5—source data 1*).

We next examined this group of expression changes for DLK-dependence, finding that about two-thirds were significantly reduced in the presence of DLKi (*Figure 5c* and *Figure 5—source data 1*). Further analysis revealed that the DLK-dependence is particularly enriched among upregulated RNAs, with over 80% of the 581 upregulated RNAs exhibiting DLKi sensitivity that reached statistical significance (*Figure 5c*). Together these findings argue that DLK-mediated stress signaling following NGF deprivation contributes more prominently to induction of stress responsive genes than to downregulation of neuronal gene expression (*e.g.*, the transcription factor *Pou4f2*), distinct from the influential role of DLK in both up- and down-regulated mRNAs previously observed in ONC (*Watkins et al., 2013*).

We next asked what portion of the DLK-mediated expression changes are dependent on the ISR. Approximately half of the DLKi-sensitive expression changes exhibit ISRIB sensitivity (*Figure 5c*), indicating the ISR is a major component of the DLK-mediated stress response, including pro-apoptotic genes (*e.g.*, *Bbc3*, which encodes the protein PUMA) (*Simon et al., 2016*) (*Figure 5d*). Importantly, however, a substantial number of DLK-mediated expression changes exhibit little to no ISR-dependence, including additional apoptotic regulators, such as *Bcl2l11*, which encodes the protein Bim, (*Biswas et al., 2007*) and Dusp16 (*Maor-Nof et al., 2016*) (*Figure 5d* and *Figure 5—source data 1*). Conversely, we evaluated the portion of ISR-mediated expression changes that exhibit DLK-dependence. Of the approximately 50% of upregulated mRNAs suppressed by ISRIB, 85% also exhibit statistically significant DLKi sensitivity (*Figure 5c*). These estimates of dual sensitivity are consistent with substantial control of the ISR by DLK.

## The ISR contributes to neurodegeneration in vitro

Given that the ISR controls a significant portion of the DLK-dependent genes induced by NGF withdrawal, we next asked whether the ISR also contributes to the DLK-mediated neuronal degeneration in this system (*Sengupta Ghosh et al., 2011*). We first scored axon degeneration in the presence of ISRIB by assessing the integrity of neurites extending from DRG explants, as this represents, at least in part, somatic pro-apoptotic signaling (*Simon et al., 2016*). We observed that, similar to inhibition of DLK (GNE-3511, 100 nM) (*Patel et al., 2015*) and JNK (JNKi, AS601245, 1 µM) (*Carboni et al.,*

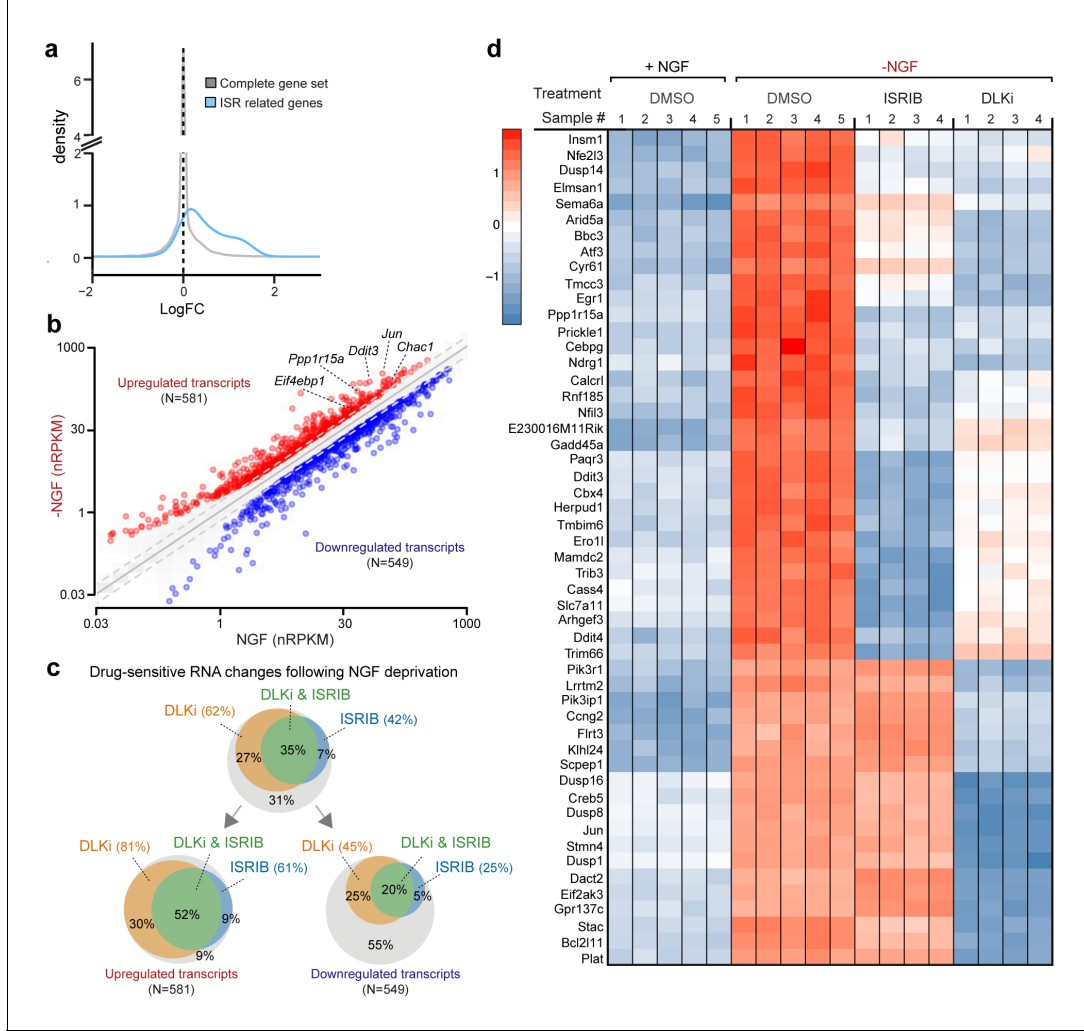

**Figure 5.** RNA-seq reveals ISRIB- and DLK inhibitor-sensitive expression changes following NGF deprivation. (**a–b**) Global expression analysis indicates an enrichment in ISR-related genes upregulated 4.5 h after NGF withdrawal from embryonic DRG cultures in the presence of DMSO vehicle (n = 5/ condition). (**a**) Density plot showing 'ISR-related' genes (blue, see Materials and methods) are more frequently upregulated compared to the distribution of all mRNAs expression changes ('complete gene set') ($p < 1 \times 10^{-5}$, one-tailed Student-t test). (**b**) Scatterplot of gene expression levels (nRPKM) in NGF-containing and NGF-deprived samples. Transcripts of ATF4 target genes *Chac1*, *Eif4ebp1*, *Ddit3*, and *Ppp1r15a* are among 581 robustly upregulated (red) RNAs following NGF deprivation (*n* = 5 per condition, >1.5 fold, adjusted-p<0.001, nRPKM = Reads Per Kilobase per Million mapped reads) (**c**) Venn diagrams reveal the portions of these NGF-responsive genes that exhibit DLKi (GNE-3511, 400 nM) and/or ISRIB (400 nM) sensitivity (see Materials and methods). (**d**) ISRIB-sensitive and ISRIB-insensitive RNAs are among those represented in a heat map of top 50 transcripts strongly upregulated in a DLKi-sensitive manner following NGF withdrawal.

The following source data and figure supplements are available for figure 5:

**Source data 1.** RNA-seq analysis of primary sensory neurons following NGF deprivation, in the presence of ISRIB or DLK inhibitor GNE-3511.

**Figure supplement 1.** Cross-platform analysis of stress-regulated mRNAs following SNC or NGF withdrawal identifies multiple ISR-associated genes (blue), including *Eif4ebp1*, *Ppp1r15a, Ddit3,* and *Chac1*, upregulated by both insults.

**Figure supplement 2.** Cross-platform analysis of stress-regulated mRNAs following ONC or NGF withdrawal identifies multiple ISR-associated genes (blue), including *Eif4ebp1*, *Ppp1r15a, Ddit3* , and *Chac1*, upregulated by both insults.

*2004*), ISRIB treatment is sufficient to reduce degeneration at 16 h (*Figure 6a,b*). However, in contrast to DLK/JNK inhibition, this protection is not sustained at 20 h (*Figure 6c,d*). We further found that axonal protection by ISRIB is mirrored by a delay in the apoptosis marker activated Caspase-3 in NGF-deprived dissociated DRG cultures, with a reduction that is similar to DLKi at 7 h but is less pronounced after 8 h (*Figure 6e,f*). These data provide evidence that ISR activity contributes to neurodegeneration, though, consistent with the ISR controlling only a subset of expression changes downstream of retrograde DLK/JNK signaling, its disruption is less protective than inhibiting DLK/JNK in this rapidly degenerating in vitro model.

## DLK- and PERK-dependent activation of the ISR following optic nerve crush

Together, our data in the NGF deprivation and SNC models suggest a pathway in which DLK/JNK signaling leads to PERK activation that drives the ISR to upregulate ATF4 and its target genes, contributing to neuronal apoptosis. To further test this hypothesis and determine its applicability to

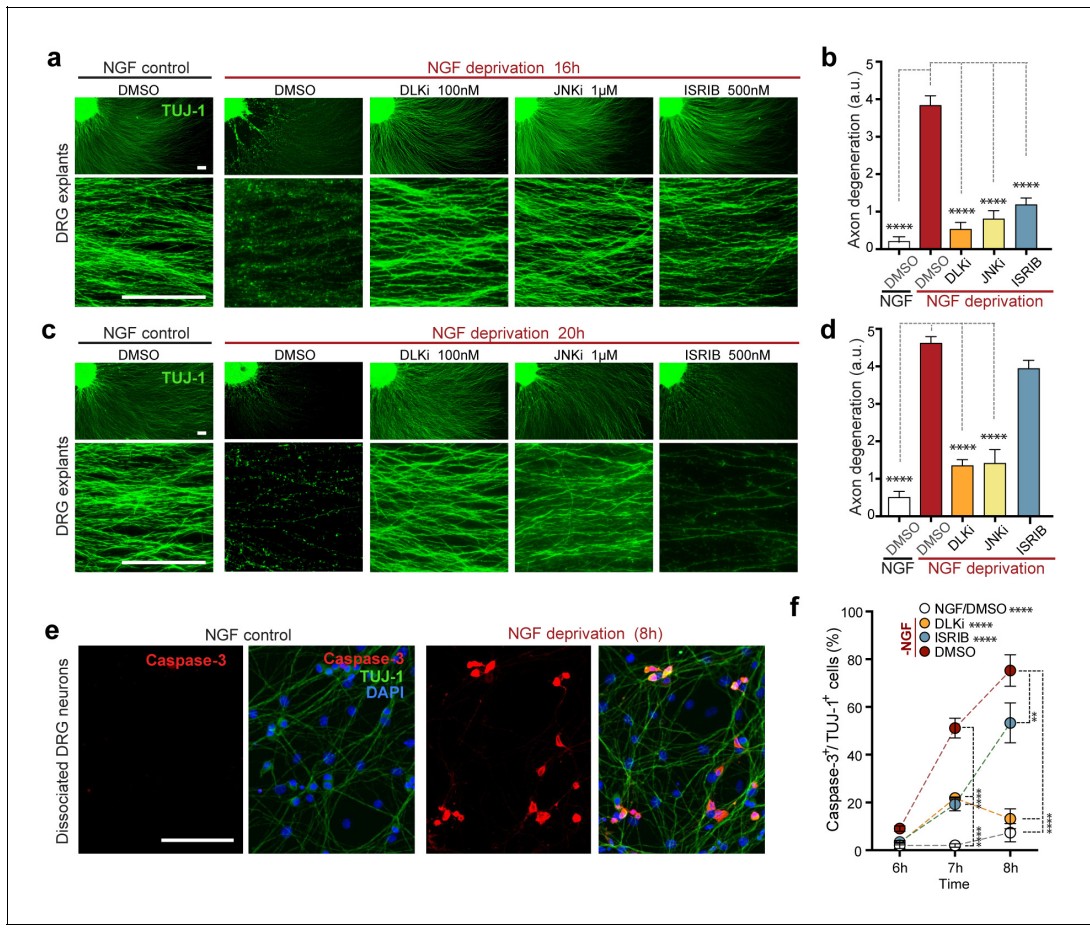

**Figure 6.** Inhibition of the ISR delays neurodegeneration following NGF withdrawal. (a–b) NGF-deprived e12.5 mouse DRG explants exhibit reduced axon degeneration (TUJ-1, green) after 16 h NGF withdrawal in the presence of ISR or JNK pathway inhibitors [ISRIB (500 nM), DLKi (GNE-3511, 100 nM) and JNKi (AS601245, 1 μM)] compared to vehicle (DMSO). (n = 8–18 explants/condition, three independent experiments). (c–d) Following 20 h of NGF deprivation, the protective effect with ISRIB is lost, in contrast to JNK pathway inhibitors. (n = 9–13 explants, two independent experiments). Blinded scoring of axon degeneration from 0–5. (e) Representative images of cleaved Caspase-3 (red) immunolabeling following NGF deprivation (8 h) in dissociated DRG cultures (TUJ-1, green), indicative of apoptosis. (f) ISRIB treatment (blue dots) reduces the proportion of DRG neurons exhibiting activated Caspase-3 immunolabeling 6–8 h following NGF withdrawal compared to vehicle control (DMSO, red dots), though not to the same extent as DLKi (orange dots) (n = 4–8 samples/condition). Scale bars = 25 μm. Data are mean ± SEM. One-way ANOVA with Bonferroni's post-hoc test was used for statistical analysis. In f, statistical significance over time for each treatment compared to –NGF/DMSO, is presented next to the condition legend and statistical significance for individual time points presented in the graph. **$p < 0.01$, ****$p < 0.0001$.

injury-induced neurodegeneration in the adult CNS, we next assessed whether this pathway is also engaged following ONC, focusing primarily on *Chac1* mRNA induction as a robust readout of pathway activation.

First, we examined the regulation of *Chac1* and other putative ATF4 target genes by qPCR. Consistent with our data in other models, genetic disruption DLK resulted in attenuation of both ATF4 protein induction (*Figure 7a* and *Figure 7—figure supplement 1*) and *Chac1*, *Eif4ebp1*, *Ddit3*, and *Ppp1r15a* mRNA upregulation (*Figure 7b*). Mice lacking JNK2 and JNK3 also lack induction of these mRNAs (*Figure 7c*), further supporting the notion that DLK/JNK signaling is necessary for ISR activation following ONC, as it is following SNC or NGF deprivation.

To definitively assess the role of neuronal PERK in transcriptional regulation and neurodegeneration following ONC, we generated a transgenic mouse line in which *Eif2ak3* is deleted in RGCs by expression of Cre recombinase under the control of the *Syn1* promoter (*Zhang et al., 2002*; *Zhu et al., 2001*). Utilizing a *R26^{LSL.tdTomato}* Cre-dependent reporter mouse, we validated that *Syn1-Cre* mediates recombination in *floxed* alleles in RGCs (*Figure 7—figure supplement 2*). Examination of *Syn1*-Cre/*Eif2ak3^{lx/lx}* retinas three days after nerve crush injury revealed a clear suppression of *Chac1* mRNA induction (*Figure 7d*), arguing that neuronal PERK mediates the DLK-dependent ISR.

To further validate the role of the ISR in gene expression changes following ONC, we examined *Chac1* expression in ISRIB-treated mice or mice in which Atf4 was targeted for knockdown in RGCs. We found that, as in SNC, systemic ISRIB dosing resulted in a significant reduction in ATF4 protein induction in retina following ONC (*Figure 7e* and *Figure 7—figure supplement 3*), as well as reduced *Chac1* mRNA upregulation (*Figure 7f*). Similarly, intravitreal injection of an AAV2 vector driving expression of both GFP and an shRNA targeting Atf4 partially suppressed *Chac1* mRNA induction (*Figure 7g* and *Figure 7—figure supplement 4*) but not *Jun* mRNA upregulation. It cannot be determined from these data whether the partial abrogation of *Chac1* mRNA reflects incomplete disruption of the ISR or additional mechanisms of *Chac1* regulation. Nevertheless, these results, along with the findings from genetic disruption of DLK, JNK2/3, and PERK, suggest that our findings regarding this pathway in distressed embryonic and adult peripheral neurons are also relevant in acute CNS axonal injury.

## PERK signaling contributes to neurodegeneration after axon injury

Given that the ISR has recently been implicated in RGC apoptosis following ONC (*Yang et al., 2016*), a model for which disruption of DLK is strongly neuroprotective (*Watkins et al., 2013*; *Welsbie et al., 2013*), we evaluated the contribution of PERK to DLK-mediated neurodegeneration following ONC. Assessment RGC survival two weeks after ONC in *Syn1*-Cre/*Eif2ak3^{lx/lx}* mice (*Figure 8a*) revealed that neuronal PERK deficiency provides partial but significant neuroprotection, when assessed either by blind scoring of axonal density within the retina (*Figure 8b*) or by automated quantitative assessment of RGC survival, with the proportion of remaining γ-synuclein[+], βIII-tubulin (TuJ1)[+] RGCs increasing from 42% to 72% (*Figure 8b*). Additionally, although microphthalmia in ATF4-deficient mice precluded ONC in that genetic model (*Masuoka and Townes, 2002*), we found that daily dosing with ISRIB (10 mg/kg) for ten days (*Figure 8—figure supplement 1*) can also provide partial neuroprotection in this model (*Figure 8c,d*). Taken together, our current findings argue that the PERK-mediated ISR cell-autonomously contributes to DLK-mediated neurodegenerative stress responses following acute neuronal insults.

## Discussion

JNK signaling and the ISR are increasingly appreciated to be important drivers of the response to distinct types of neuronal insult. The observations reported here reveal that following acute neuronal injury, these seemingly distinct cellular stress response pathways are tightly linked via the neuronal MAP kinase kinase kinase DLK. We found that the ISR-activating kinase PERK contributes to DLK-mediated gene expression changes and neurodegeneration. Thus, activation of PERK represents a second functional component, along with JNK signaling, of the DLK-driven stress response that regulates the fates of distressed neurons (*Figure 9*). Evidence supporting this hypothesis includes the observations that acute insults known to stimulate DLK stabilization also engage the ISR through PERK, that DLK deficiency prevents ISR activation following these insults, and that DLK activity is



**Figure 7.** DLK/PERK signaling regulates ATF4 target gene *Chac1* mRNA following optic nerve crush. (**a–b**) DLK-deficient (*CAG-ERT^pos^;Map3k12^lx/lx^*, DLK cKO) mice exhibit reduced activation of the ISR and JNK pathway following ONC compared to controls (*CAG-ERT^neg^;Map3k12^lx/lx^*). (**a**) DLK-deficient retina lysates 3 d after ONC display no induction of ATF4 protein by immunoblot (n = 4/condition). (**b**) qRT-PCR of ISR-associated genes in retina following ONC in DLK cKO mice compared to Cre-negative controls (n = 5–6/ condition). (**c**) qRT-PCR of ISR-associated genes in retina following ONC (red) in JNK2/3-deficient mice (n = 5–6/condition). (**d**) Induction of *Chac1* mRNA is suppressed in PERK cKO retinas (*Syn1-Cre;Eif2ak3^lx/lx^* mice) compared to Cre-negative littermates following ONC (n = 6). (**e**) Dosing with ISRIB reduces upregulation of ATF4 protein in mouse retinal lysates 16 h post-ONC (immunoblot and quantification, n = 4/condition, 2.5 mg/kg ISRIB b.i.d.). (**f**) *Chac1* mRNA (qPCR, n = 4/condition) in mouse retina 48 h post-ONC (10 mg/kg ISRIB b.i.d.). (**g**) Intravitreal injection of an AAV2 vector directing expression of GFP and an shRNA targeting mouse *Atf4, GFP-shRNA (Atf4),* reduces *Chac1* mRNA induction following ONC in transduced regions of the retina compared to a similar AAV2 vector with a non-targeting shRNA sequence, *GFP-shRNA(Ctrl)* (see Materials and methods, qPCR, n = 8/condition). One-way ANOVA with Bonferroni's post-hoc test was used for statistical analysis. Data are mean ± SEM. *p<0.05, **p<0.01, ***p<0.001, ****p<0.0001.

The following figure supplements are available for figure 7:

**Figure supplement 1.** Quantification of p-c-Jun immunoblots presented in *Figure 7a* (DLK-deficient retina lysates three days after ONC, n = 4/condition).

**Figure supplement 2.** Syn1-Cre activity in the retina, illustrated by *Syn1-Cre;R26^LSL.tdTomato^* (blue) expression co-localizing with TUJ-1 (green) and γ-synuclein (red) immunolabeled RGCs in the GCL.

**Figure supplement 3.** Quantification of p-c-Jun immunoblots presented in *Figure 7e* (ONC in mice dosed with ISRIB).

**Figure supplement 4.** qPCR of *Jun* mRNA following ONC in AAV2-shRNA-GFP-injected mice (corresponding to *Figure 7g*).

sufficient to drive PERK phosphorylation and ATF4 upregulation in non-neuronal cells. Furthermore, our RNA-seq data suggest that greater than 80% of ISRIB-sensitive expression changes are also DLKi-sensitive. This estimate suggests a remarkable degree of overlap, given that it does not account for the many potential differences in the selectivity, metabolism, and stability of DLKi and

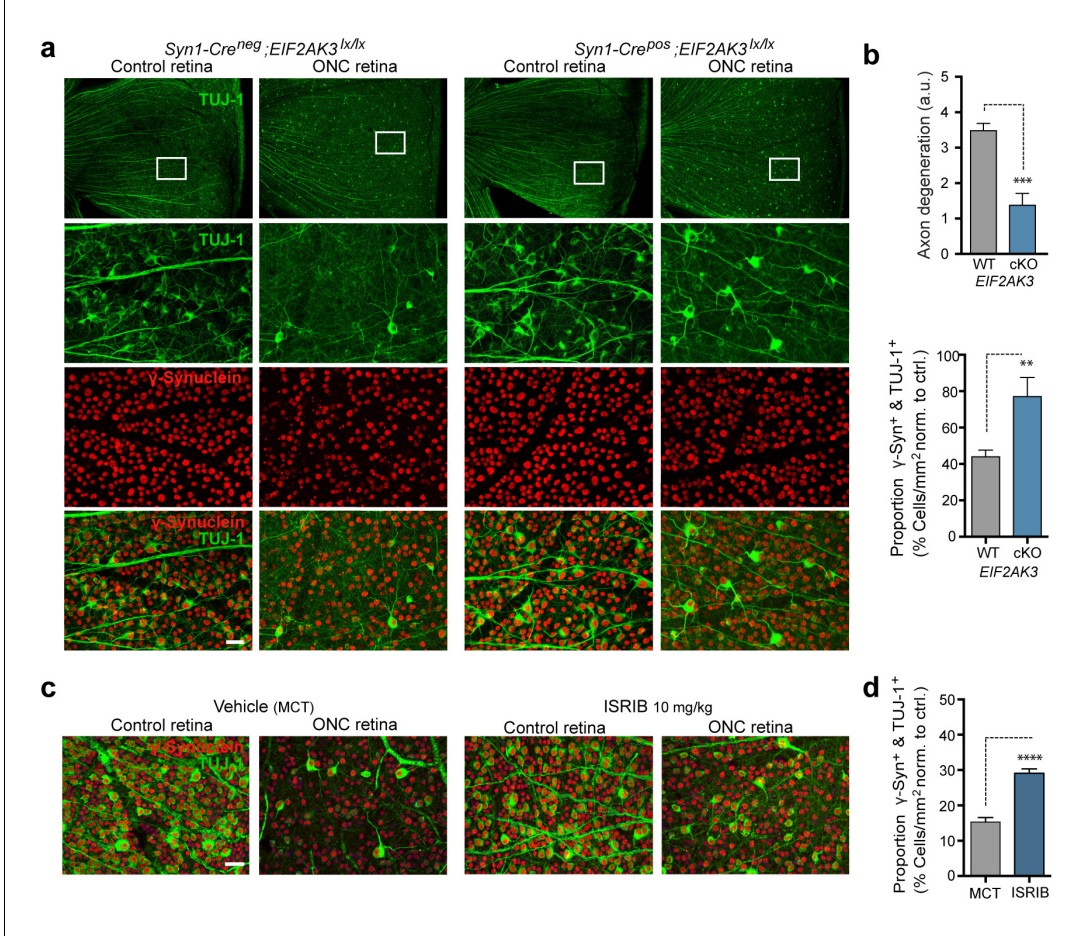

**Figure 8.** PERK contributes to neurodegeneration following nerve injury. (**a–b**) Reduced loss of PERK-deficient RGCs 14 d after ONC. (**a**) TUJ-1 (green)/ γ-synuclein (red) immunolabeled whole mount retinas from control *Syn1-Cre*[neg];*Eif2ak3*[lx/lx] (PERK WT) and neuronal PERK-deficient *Syn1-Cre*[pos]; *Eif2ak3*[lx/lx] (PERK cKO) mice 14 d after ONC, high magnification (expanded from boxed region). (**b**) Blinded scoring of axon degeneration from 0–5 in whole-mounted retinas and automated image quantification of TUJ-1/γ-synuclein double-immunolabeled RGCs normalized to uncrushed retina of each mouse (n = 8 WT, n = 6 PERK cKO). (**c–d**) Reduced loss of RGCs 10 d post-ONC in ISRIB-treated mice (10 mg/kg ISRIB in MCT vehicle b.i.d) compared to vehicle control. (**c**) Automated image analysis of TUJ-1 (green, outlined in cyan)/ γ-synuclein (red, outlined in blue) immunolabeled cells. (**d**) Quantification of TUJ-1/γ-synuclein double-immunolabeled RGCs (outlined in green) in crushed retinas normalized to uncrushed control (n = 8). Scale bars = 25 μm. Student's *t*-test was used for statistical analysis. Data are mean ± SEM. \*\*p<0.01, \*\*\*p<0.001, \*\*\*\*p<0.0001.

The following figure supplement is available for figure 8:

**Figure supplement 1.** Pharmacokinetic analysis in mice dosed IP with ISRIB formations in MCT (red) and nanosuspenion in MCT (blue) over one day, as determined by plasma concentration (left).

ISRIB or differences in basal ISR and DLK pathway activity, both of which may influence the relative sensitivity of each transcript to each compound. c-Jun deficiency (*Fernandes et al., 2012*) and PERK deficiency are each partially neuroprotective after ONC, and the control of both of these pro-apoptotic signaling pathways by DLK is consistent with the potent neuroprotection provided by DLK disruption (*Watkins et al., 2013*; *Welsbie et al., 2013*).

How does DLK direct PERK activation? It is possible that extensive DLK-driven gene expression overwhelms the protein folding machinery resulting in ER stress. However, a distinct PERK-independent branch of the UPR is only transiently activated after ONC (*Hu et al., 2012*), and protein synthesis inhibitors fail to prevent PERK activation during neuronal ischemia (*Sanderson et al., 2010*). Our findings from Campenot chamber experiments suggests that PERK activation occurs downstream of DLK/JNK-mediated retrograde injury signaling, though additional experiments suggest that it is not

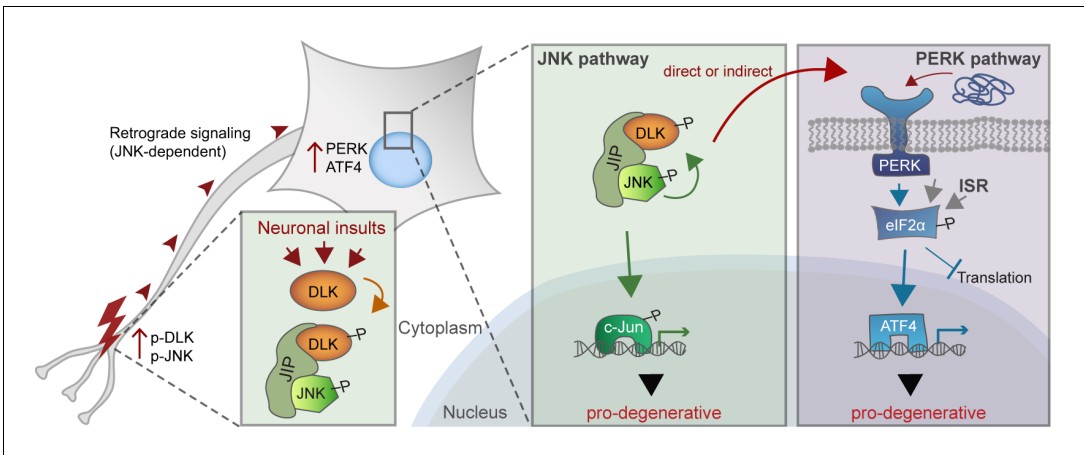

**Figure 9.** A working model of DLK as a master regulator of multiple neuronal stress response pathways. DLK (orange) regulates apoptotic transcriptional changes through both JNK signaling (green) and the PERK-driven ISR (blue) (which can also be activated by misfolded proteins in the ER). One potential mechanism of cross-talk may be the activation of PERK by DLK activity (either directly or indirectly) following retrograde signaling mediated by a complex that includes DLK, JNK, and the JNK-interacting protein (JIP).

dependent on c-Jun. Perhaps DLK or one of its downstream effectors directly or indirectly mediates phosphorylation (or other post-translational modification) events of PERK or PERK regulators to initiate the ISR (*Figure 9*). Further studies may therefore uncover novel molecular interactions between the UPR and MAP kinase signaling components (*Urano et al., 2000*).

Our findings reveal that PERK signaling, which is initiated by ER stress as one branch of the UPR, can also be triggered by the MAP kinase DLK following acute neuronal insults. This raises the question of whether regulation of the ISR is a general feature of neuronal MAP kinase stress signaling. Our observation that DLK activation is sufficient to activate PERK suggests the possibility that activation of the ISR may be a common consequence of MAP kinase stress signaling in DLK-expressing cells, such as neurons. A number of recent studies implicate the ISR in models of neurodegenerative conditions, with beneficial effects of PERK and ISR inhibition that are tempered by concerns of pancreatic toxicity, yet it remains to be determined whether ER stress and the classical UPR account for PERK activation in these settings (*Halliday et al., 2015*; *Moreno et al., 2013*). Although the precise role(s) of DLK in neurodegenerative diseases requires further investigation, JNK activation has been noted in many models of progressive neurodegeneration (*Coffey, 2014*). It is therefore possible that DLK activity influences or mediates PERK signaling in these settings. Given the neuronally restricted expression of DLK (*Hirai et al., 2006*) and the viability of adult DLK-deficient mice (*Watkins et al., 2013*), blocking DLK activity may provide an appealing therapeutic strategy for simultaneously targeting pro-degenerative PERK and JNK signaling in neurons.

## Materials and methods

### Mouse models

DLK-deficient mice were generated as previously described (*Watkins et al., 2013*), utilizing tamoxifen feeding of *CAG^ERT-Cre* (Jackson Laboratory) mice crossed to *Map3k12^lx/lx* mice (*CAG-ERT^pos*; *Map3k12^lx/lx* referred to as DLK cKO, RRID:MGI:5502227). *Map3k12^lx/lx* mice lacking Cre but treated with tamoxifen were used as controls (*CAG-ERT^neg*; *Map3k12^lx/lx* referred to as DLK WT RRID:MGI: 5294552).

*Mapk9* (RRID:MGI:3619033) *and Mapk10* (RRID:MGI:5551975) double knockout mice were generated as previously described (*Huntwork-Rodriguez et al., 2013*). JNK2/3-deficient mice (*Mapk9^{−/−}*; *Mapk10^{−/−}*) were compared to control mice (carrying both or at least one WT allele of *Mapk9* and *Mapk10*).

Mice harboring a null allele of *Atf4* were originally generated with a conditional *loxp* knockout cassette (*Yoshizawa et al., 2009*). These mice (RRID:MGI:4360510), however, exhibited phenotypes (microphthalmia, reduced body weight, low survival rate to adulthood) characteristic of *Atf4* knockout mice (*Masuoka and Townes, 2002*) in the absence of Cre (*data not shown*), and further sequencing analysis revealed that the insertion of the *loxp* cassette disrupted the first splice site and resulted in an allele that, in contrast to the wildtype allele, did not result in expression of ATF4 protein after SNC (*Figure 2—figure supplement 2*).

Neuronal PERK deficient mice were generated by crossing mice expressing Cre under the *Syn1* promoter RRID:MGI:2176767 (*Zhu et al., 2001*) with *Eif2ak3$^{lx/lx}$* mice RRID:MGI:3618624 (*Zhang et al., 2002*) (*Syn1-Cre$^{pos}$;Eif2ak3$^{lx/lx}$*, PERK cKO), Cre-negative mice served as controls (*Syn1-Cre$^{neg}$;Eif2ak3$^{lx/lx}$*, PERK WT). For assessment of the types of neurons that express Cre and evaluation of regeneration of motor and large-diameter sensory axons after SNC in vivo, mice expressing *Syn1-Cre* and a *lox-STOP-lox (LSL)* tdTomato Cre reporter allele (*Gt(ROSA)26Sor$^{tm14(CAG-tdTomato)Hze}$*, RRID:IMSR_JAX:007914, referred to as *Syn1-Cre;R26$^{tdTomato}$*, Allen Brain Institute) were used as controls for mice harboring those same alleles in addition to *Eif2ak3$^{lx/lx}$*.

## Antibodies and inhibitors

The following antibodies were used: ATF4 (1:800, rb, Cell Signaling Technology, RRID:AB_2616025), p-c-Jun serine 63 (1:250, rb, Cell Signaling Technology, RRID:AB_2130162), anti-cleaved-Caspase-3 (1:400, Cell Signaling #9661, RRID:AB_2341188), DLK (1:1000, rb, Genentech), eIF2α (1:1000, Cell Signaling #9722, RRID:AB_2230924), p-eIF2α (1:1000, rb, Cell Signaling Technology, RRID:AB_2096481), PERK (1:1000, rb, Cell Signaling Technology, RRID:AB_2095847), p-PERK (1:250, rb, Cell Signaling Technology, RRID:AB_2095850), γ-synuclein (1:200, ms, Abcam, RRID:AB_2193398), TUJ-1 (1:1000, rb, Covance/Biolegend, RRID:AB_2566588), TUJ1 (1:1.000, ms, MMS-435P, Covance/Biolegend, RRID:AB_2313773), TUJ-1–488 (1:500, rb, Covance/Biolegend, RRID:AB_2564757), GAPDH-HRP (1:1000, Cell Signaling Technology, RRID:AB_1642205). Note that the anti-p-PERK antibody recognizes mouse but not human phospho-PERK, which we detected based on a shift in mobility when utilizing the anti-PERK antibody. Secondary antibodies used for immunofluorescence were Alexa 488, 568, 594, 647 (1:400, Jackson Immunoresearch Laboratories) and for immunoblot detection anti-rabbit HRP conjugated (1:10000, VWR International).

The following inhibitors were used: T9033 (Thapsigargin, SIGMA), ISRIB (*Sidrauski et al., 2013*), GNE-3511 (DLKi) (*Patel et al., 2015*), and AS601245 (JNKi). For concentrations and dosing, see below.

## In vivo nerve injury models

Sciatic nerve crush (SNC) experiments were performed essentially as described (*Zou et al., 2009*). In brief, mice were anesthetized using isoflurane, and the sciatic nerve was exposed and crushed for 10 s using fine-tip forceps medial to the sural nerve branch. After 1–3 days, the associated L4 DRG or combined L3 and L4 DRG (*Rigaud et al., 2008*), as well as the contralateral uninjured DRGs as controls, were harvested and processed for qRT-PCR, immunoblot and immunohistochemistry analysis (see details below). Assessment of axon regeneration in tdTomato-expressing axons (with and without *Syn1-Cre*-mediated deletion of *Eif2ak3*) was performed following perfusion fixation seven days after SNC. Whole-mounted nerves were imaged by epifluorescence, and the crush site (indicated by depressed tdTomato brightness) and regeneration front (indicated by the end of the majority of tdTomato signal and confirmed by higher magnification two-photon imaging, as shown in *Figure 1—figure supplement 5*) were blindly identified for measurement of regeneration distance. Sciatic nerve biopsies, 5–7 mm in close proximity to the SNC injury site (upstream of the SNC site, distal to the muscle), were collected for immunoblot analysis.

Optic nerve crush (ONC) was performed as described (*Watkins et al., 2013*). The left eye of each animal was subjected to optic nerve injury via an intraorbital incision made using a fine-tip forcep 1–2 mm from the eyeball. After 1–3 days, retinas were harvested and processed for qRT-PCR or immunoblot analysis (see below). Assessment of RGC survival in fixed, whole-mounted retinas after 14 days was performed as described (*Watkins et al., 2013*), with the addition of a second marker (TUJ-1) to allow for automated identification and quantification of TUJ1- and γ-synuclein-expressing RGCs

specifically rather than simply the γ-synuclein-expressing ganglion cell layer (GCL) neurons that include displaced amacrine cells.

## ISRIB dosing in adult mice

For ONC and SNC immunoblot experiments, ISRIB was dosed at 2.5 mg/kg b.i.d. in 50% DMSO, 50% PEG400, and for qPCR analysis and for ONC neuroprotection analysis, ISRIB was dosed at 10 mg/kg in medium chain triglycerides (MCT) nanosupension b.i.d. Pharmacokinetic parameters for ISRIB were calculated in blood plasma for 2.5, 5, and 10 mg/kg ISRIB at various time points. ISRIB nanosuspensions in MCT offered the best PK properties, with 10 mg/kg resulting in ca 4x higher plasma concentration compared to 2.5 mg/kg (*Figure 8—figure supplement 1*). Retinas and L3/L4 DRG were processed for qRT-PCR, immunoblot, and immunohistochemistry analysis (detailed below).

## Intravitreal injection of adeno-associated viral (AAV) vectors

Recombinant AAV serotype 2 (AAV2) viral vectors were produced by SignaGen. The vector targeting mouse *Atf4* for knockdown included an shRNA with a sequence, 5'-GCGAGTGTAAGGAGC TAGAAA-3', previously validated in vivo (*Matus et al., 2013*) under the control of the U6 promoter, as well as GFP under the control of the CMV-$\beta$-actin-globin (CAG) promoter. Control vector included a non-targeting shRNA scramble sequence. Intravitreal injection was performed as previously described (*Watkins et al., 2013*), using 3 µl of vector at $1 \times 10^9$ vector genomes per µl. ONC was conducted seven days post-injection and retinas were harvested four days post-injury. GFP expression in freshly isolated retinas was variable, suggesting incomplete transduction and partial *Atf4* targeting. Therefore, only retinal regions with prominent GFP-expression were dissected and lysed for expression analysis.

## Assessment of RGC survival

Degeneration of RGCs after ONC were imaged and quantified automatically. Whole mounted retinas were scanned with a Nanozoomer XR automated slide scanning platform (Hamamatsu, Hamamatsu City, Shizuoka Pref., Japan) at 200x final magnification. Scanned slides were analyzed using Matlab (version R2013a by Mathworks, Natick, MA). Standard morphological filters were used to define regions approximately 150 µm distance from the edge on the periphery of each leaf of the whole-mount retina with a thickness of approximately 500 µm. Individual RGC nuclei within these regions were identified using an algorithm based on radial symmetry (*Veta et al., 2013*) applied to the γ-synuclein channel. A ring-shaped region was created for each cell, centered on the border of the nucleus. Positive cells were defined as having greater than 50% of this ring-shaped region classified as positive for TUJ-1 staining as determined by Otsu's threshold method (*Otsu, 1979*). Large contiguous regions of TUJ-1 staining associated with axonal bundles were identified using standard morphological filters and removed from consideration when scoring individual cells. The degree of retinal axon degeneration 14 days after ONC was assessed from in whole-mount retinas by blinded scoring of density of axon bundles immunolabelled with TUJ-1, using a graded scale from 0–5, where 0 corresponds to no degeneration, and five to complete degeneration.

## Primary neuron culture and NGF withdrawal

DRGs were dissected from E12.5 to E13.5 CD-1, or *Map3k12* KO mouse embryos, and cultured as described previously (*Sengupta Ghosh et al., 2011*; *Huntwork-Rodriguez et al., 2013*). In brief, dissociated DRG neurons and DRG explants were grown in F12 medium containing N3 supplement, 40 mM glucose, and 25 ng/ml NGF and grown in poly-D-lysine (PDL)- and laminin-coated wells. Cytosine arabinofuranoside (AraC; 3 µM, Sigma-Aldrich) was added to the medium 24 hr after plating and removed 24 hr later.

For NGF-deprivation experiments, culture medium was replaced with medium without NGF and with 50 µg/ml anti-NGF antibody (Genentech) after 3–4 days in vitro. For inhibitor treatment experiments, DRGs were pre-incubated for 30–60 min with media supplemented with vehicle (DMSO) or inhibitors in DMSO; ISRIB (50, 100 or 500 nM), DLKi (GNE-3511; 50, 100, 250 nM or 500 nM) and JNKi (AS601245, 1 and 10 µM). Thapsigargin (100 nM) was used as a positive reference control of ER-stress and added with NGF-containing media. For immunoblot experiments, NGF was withdrawn

for 3 hr. For axon degeneration assay, NGF was withdrawn for 16 or 20 hr, explants were fixed and immunolabeled for TUJ-1. For qPCR and RNA sequencing, NGF was withdrawn from dissociated primary DRGs for 4.5 hr. For apoptosis experiments, NGF was withdrawn from dissociated DRGs for 6–8 hr, fixed and immunostained for (active) cleaved-Caspase-3 and TUJ-1.

siRNA-mediated knockdown of gene expression in embryonic DRGs was performed using Accell siRNA SMARTpools (Dharmacon): *Jun* (E-043776-00-0005), *Eif2ak1* (E-045523-00-0005), *Eif2ak2* (E-040807-00-0005), *Eif2ak3* (E-044901-00-0005) and *Eif2ak4* (E-044353-00-0005), Dharmacon siControl RNA was used as a transfection control. Accell siRNAs were added together with AraC in F12/N3 cell culture medium 20 hr after plating, and replaced with new culture medium 24 hr later. NGF withdrawal was performed following 3–4 days in culture.

Axon degeneration after NGF-withdrawal was quantified as previously described (*Sengupta Ghosh et al., 2011*). In brief, DRG explant axons were imaged at 20x magnification, taken at similar distance from explant center. Axon degeneration was scored blindly on a scale from 0–5, in which 0 equals intact axons and five equals complete axon degeneration (8–18 explants per condition). Apoptosis, induced by NGF withdrawal, was measured by quantifying the proportion of Caspase-3 immunopositive TUJ-1 marked DRG neurons (n = 4–10 wells/condition, the average proportion (%) of 2–3 images per well were quantified blindly and used for statistical comparison).

## Microarray analysis

Microarray analysis was performed on whole retinas 3 days after ONC and L3/L4 DRGs 3 days after sciatic nerve transection in adult mice using uncrushed retinas and DRGs as controls, as previously described (*Watkins et al., 2013*).

Retinas and DRGs were dissected and immediately transferred to RNAlater (Qiagen) and RNA isolated using RNeasy Plus Mini Kit (Qiagen). RNA sample quantity and quality were determined using an ND-1000 spectrophotometer (Thermo Scientific) and Bioanalyzer 2100 (Agilent Technologies), respectively. Cyanine (Cy)-dye labeled cRNA were hybridized to a Whole Mouse Genome 4 × 44Kv2 arrays and scanned with an Agilent microarray scanner. Statistical analyses were performed using R as previously described (*Watkins et al., 2013*). For determination of enrichment of after nerve injury, 31 'ISR-related' genes were defined as mRNAs by their reduction in cortical neurons from ATF4$^{-/-}$ mice and by their upregulation upon neuronal oxidative stress in an ATF4-dependent manner, according to a published expression analysis (*Lange et al., 2008*).

Gene set analysis was used to test for enrichment of the 'ISR-related' genes after SNC and ONC (data shown in *Figure 1—figure supplementary 1*). This analysis was performed using the roast function in the limma package with number of rotations set to 1,000,000.

## Quantitative RT-PCR (qPCR) analysis

RNA was isolated from primary DRG neurons, adult L3/L4 DRGs and retinas using RNeasy Plus Mini Kit (Qiagen). For primary DRG experiments NGF was withdrawn for 4.5 hr in the presence of DMSO, DLKi or ISRIB. qPCR was performed under standard conditions using the TaqMan RNA-to-Ct One-Step kit (Applied Biosystems) on a 7500 Real Time PCR System (Applied Biosystems). Transcript levels were measured in triplicates, normalized to *Gapdh* (Mm03302249_g1) (VIC labeled) and compared to wild type control samples not subjected to nerve crush, or control, using the comparative Ct (*ΔΔ*Ct) method. Predesigned TaqMan primers were obtained from Applied Biosystems, including those for *Chac1* (Prod No. Mm00509926_m1), *Ddit3* (Prod No. Mm01135937_g1), *Ppp1r15a* (Prod No. Mm01205601_g1), *Eif4ebp1* (Prod No. Mm04207378_g1) and *Gapdh* (Prod No. Mm03302249_g1).

## High-throughput RNA sequencing

RNA sequencing was conducted for primary NGF-deprived DRG neurons (4.5 hr) in the presence of vehicle (DMSO), 400 μM DLKi, or 400 μM ISRIB. RNA was isolated from primary DRGs grown for four days in vitro (ca 400,000 cells/well) treated with NGF/DMSO, anti-NGF/DMSO, ISRIB/anti-NGF, and DLKi/anti-NGF, using RNeasy Plus Mini Kit (Qiagen). RNA samples were sequenced as previously described (*Srinivasan et al., 2016*). 1 μg of total RNA was used as an input material for library preparation using TruSeq RNA Sample Preparation Kit v2 (Illumina). Libraries were multiplexed and

sequenced on Illumina HiSeq2500 (Illumina) to generate on average 30 million 50 base pair single end reads per library.

## RNA-Seq alignment

The fastq sequence files for all RNA-seq samples were filtered for read quality and ribosomal RNA contamination. The remaining reads were then aligned to the mouse reference genome (GRCn38) using the GSNAP alignment tool (*Wu and Nacu, 2010*). Alignments were produced using the following GSNAP parameters: '-M 2 n 10 -B 2 -i 1 N 1 w 200000 -E 1 –pairmax-rna = 200000 –clip-overlap'. These steps, and the downstream processing of the resulting alignments to obtain read counts and normalized Reads Per Kilobase Million (nRPKMs) per gene (over all exons of RefSeq gene models), are implemented in the Bioconductor package, HTSeqGenie (v 3.12.0) (*Pau and Reeder, 2014*). Only uniquely mapped reads were used for further analysis.

## RNA-seq differential gene expression

Read counts were adjusted based on DESeq2 estimated sizeFactor (*Love et al., 2014*) and weighted by their mean-variance relationship as estimated by voom. Differential gene expression was performed using the limma empirical Bayes analysis pipeline described in the R package limma (*Ritchie et al., 2015*). A pre-filter was applied on the read counts such that only genes with at least ten counts in at least three samples (of any condition) were analyzed. P-values for other genes were simply set to 1 and log-fold-changes to 0 for visualization purposes, but such genes were not included in the multiple testing correction.

## RNA-seq data presentation

For heatmaps, genes were log2-transformed, then scaled and zero centered using the z-transform. An arbitrary floor of $-4$ was set for genes with a log2 RPKM value of less than $-4$. Heatmaps were plotted using the superheat package in R. Venn diagrams were generated using 3-Way Venn Diagram Generator (http://jura.wi.mit.edu/bioc/tools/venn3way/). ISRIB- and/or DLKi-sensitivity was defined for 1130 strongly NGF-regulated transcripts as a significant difference (>1.25 fold, *adjusted p*<0.05) between each drug treatment condition (n = 4) and the anti-NGF ('-NGF') condition treated with vehicle (DMSO, n = 5).

## Accession number

RNA-sequencing and SNC microarray data has been deposited to the Gene Expression Omnibus (GEO) under accessions GSE95672 and GSE96051, respectively. Previously published ONC microarray data (*Watkins et al., 2013*) has been deposited to the Gene Expression Omnibus under accession GSE96053.

## Immunoblot and immunohistochemistry

Mouse eyes/retinas and DRG were dissected in ice-cold PBS, snap frozen or fixed in 4% paraformaldehyde (PFA) and cultured cells were either fixed in 4% PFA or lysed.

For immunoblot experiments of retina, DRG tissue or cultured cells samples were homogenized by sonication at 4°C in RIPA or m-PER lysis buffer supplemented with Complete protease and PhosSTOP phosphatase inhibitor cocktails (Roche). Equal amount of protein per sample were loaded onto 4–12% BisTris gels (Bio-Rad), separated in the presence of MOPS buffer and transferred to nitrocellulose membranes (Bio-Rad). Immunoblots were developed using SuperSignal Dura and Femto substate (Life Technologies) and images were captured using a ChemiDoc (Bio-Rad Laboratories).

Immunoblots were quantified using Image Lab v. 5.2.1 (Bio-rad Laboratories) or Image Studio Lite (LI-COR Biotechnology). Band intensities for each sample were normalized to loading control (GAPDH or TUJ-1). Independent experiments were further normalized to NGF-controls for statistical comparison.

Immunohistochemistry was performed on mouse retinas and DRGs. Cryo-sections (16 µm) were blocked in 2% donkey serum and 0.2% Triton X-100 (Sigma-Aldrich) in PBS and floating whole mouse retinas were blocked in 5% normal goat serum in and 0.5% Triton X-100 in PBS. Primary antibodies were incubated overnight at 4°C. The following day tissues/sections were rinsed and

incubated with secondary antibodies in PBS. Primary dissociated DRGs and DRG explants were blocked in 5% normal goat serum and 0.25% Triton X-100 in PBS for 30 min and incubated with primary antibody for 3 hr at room temperature or overnight at 4°C. Secondary antibodies were incubated for 1 hr at room temperature in PBS. Retinas were mounted on slides after being cut with four incisions along the radial axes (as illustrated in *Figure 1A*). Sections or primary cells were mounted in Fluoromount-G (SouthernBiotech).

Immunofluorescence images of DRG explants were acquired using a fluorescent microscope (DM5500, Leica) with a DFC360 camera. Retinas, cryosections and primary DRG cultures were imaged using a confocal microscope (LSM710, Carl Zeiss). Images were processed for brightness and contrast in Adobe Photoshop (Adobe) or Fiji (built on the ImageJ2 platform) (*Schindelin et al., 2012*).

## Axon isolation preparation and campenot chambers

Isolation of axons from DRG neurons were conducted as described previously (*Unsain et al., 2014*). Briefly, DRG explants (30–35/condition) were plated on top of porous cell culture filters (87717, Falcon) coated with PDL, laminin and collagen. Explants were treated with AraC and cultured for three days in F12 medium containing N3 supplement, 40 mM glucose, with 25 ng/ml NGF in the inner (somal/axonal) compartment and 50 µg/ml NGF the outer (axonal) compartment. Global NGF-deprivation was conducted for 1, 3, or 6 hr as described above. Axons attached below the porous cell culture filters were separated from explants (cell bodies and axons) and analyzed by immunoblot.

Compartmentalized (Campenot) chamber experiments were performed as previously described (*Campenot, 1977*; *Nikolaev et al., 2009*). In brief, Campenot chambers were assembled by positioning Teflon dividers (Camp8, Tyler Research) on poly-d-lysine and laminin coated 35 mm tissue culture dishes with tracks (for axonal elongation) and sealed with silicone grease. DRGs prepared from E12.5 mouse embryos were plated in the cell body compartment and grown for 3–5 days in Neurobasal methylcellulose-thickened medium (NBM-MC) supplemented with B27, L-glutamine, glucose, pen-strep (Gibco) and 50 ng/ml NGF. DMSO (control) or inhibitors diluted in DMSO were added to the culture media of either the axonal or cell body compartment 1 hr prior to NGF-deprivation. NGF was withdrawn from the axonal or cell body compartments by replacing media with NBM-MC media containing 50 µg/ml anti-NGF antibody (without NGF) for 5 hr. Following NGF-deprivation, DRGs from the cell body compartment were harvested for subsequent immunoblot analysis.

## Adult sensory neuron neurite outgrowth bioassay

For adult DRG experiments assessing neurite outgrowth in vitro, protocols were adapted from previous studies (*Smith and Skene, 1997*; *Zou et al., 2009*). Briefly, DRGs were dissected from 7–9 week old female CD-1 mice and dissociated by sequential treatment with collagenase (3 mg/ml, 90 min, 37°C, Worthington LS004196) and trypsin (0.25%, 30 min, 37°C, Sigma T74091G), followed by trituration. This dissection and dissociation axotomizes the neurons, engages the DLK-dependent upregulation of regeneration-associated genes, and represents the only 'injury' in these experiments (i.e., no in vivo or in vitro conditioning lesion models were employed). Following centrifugation through a 15% BSA (Sigma A9418) cushion to remove debris and resuspension in 0.02% BSA-containing buffer, IB4-positive cells were negatively selected by incubation for 10 min at room temperature on a Petri dish that had been coated with IB4 lectin (2 µg/ml, Sigma L3019, 4 hr, room temperature) and further blocked with 0.2% BSA (30 min room temperature) (*Kalous and Keast, 2010*). Following centrifugation and resuspension in Neurobasal-A and B27-containing medium (Gibco 35050), cells were mixed with complexes of 100 µM siRNA (OnTarget Plus SMARTpools, Dharmacon) and Dharmafect two transfection reagent (Dharmacon). Cells were then plated on 96-well plates pre-coated with PDL and further coated with laminin (2 µg/ml, 4 hr, 37°C, Invitrogen 23017–015) and Nogo/Fc (R&D Systems). After 42 hr of culture, cells were fixed with 4% paraformaldehyde, blocked with 5% BSA, and immunostained using TUJ-1-Alexa 488 (1:500; Covance). Neurite growth was assessed by automated image analysis following acquisition on a GE INCell 2000 cell analyzer. For each well, the total length of TUJ-1-positive neurites was divided by the number of TUJ1-positive cell bodies, and that value was compared to the siCONTROL non-targeting siRNA pool.

## HEK293 cell expression experiments

Human embryonic kidney 293T (HEK293T, RRID:CVCL_0063) cells were used for overexpression of human DLK (*Sengupta Ghosh et al., 2011*), mouse DLK, and a kinase dead DLK-S302A (*Huntwork-Rodriguez et al., 2013*). An empty DNA vector (pRK) was used as negative transfection control. Cells were grown to ca 75% confluence in DMEM media supplemented with 1× Glutamax (Gibco), 1× penicillin/streptomycin (Gibco) and 10% fetal bovine serum in 6-well plates. Transient transfections were performed using Lipofectamine LTX with Plus Reagent according to manufacturer's protocol (Life Technologies), whereas stable human-DLK expressing HEK293 cells (RRID:CVCL_V350) (*Sengupta Ghosh et al., 2011*) were induced by 2 µM doxycycline (DOX). Small molecule inhibitors were added together with DOX, and cells were lysed 20–24 hr later. Non-transfected control cells were given Lipofectamine reagents without plasmid DNA.

### Cell line authentication/quality control

Short Tandem Repeat (STR) profiles were determined for each line using the Promega PowerPlex 16 System. All stocks were tested for mycoplasma by two methods (Lonza Mycoalert and Stratagene Mycosensor) prior to and after cells are cryopreserved.

## Statistical analysis

Statistical significance was evaluated in Graph pad Prism v. 6.00 for Mac (GraphPad Software, www.graphpad.com). One-way ANOVA with Bonferroni's post hoc test was used where appropriate ($p < 0.05$ was considered as a significant change between groups). Statistical comparison between two groups were analyzed using two-tailed Student's t-test. For p-values and number of replicates (sample and/or experiment) used for statistical comparison see figure legends, with additional details provided in the related sections of the Methods. Sample-sizes were determined prior to each experiment and based on previously published data and preliminary experiments using the same methods (*Sengupta Ghosh et al., 2011*; *Huntwork-Rodriguez et al., 2013*; *Watkins et al., 2013*). Accordingly, no explicit power analyses were used for sample size estimation. Outliers were excluded based on Grubbs' test (Alpha = 0.05). For immunoblot quantifications after siRNA knockdown and in the Campenot chamber setup, only experiments which resulted in a robust elevation of ATF4 following anti-NGF compared to NGF-controls were included. All data are presented as means ± SEM.

# Additional information

### Competing interests

ML: Employed by Genentech. SH-R, HS: Employed by Genentech at time of study and own shares. ZJ, JSK, JE-A, MS, ZM: Employed by Genentech and own shares. KH: Employed by Genentech at the time of the study. MT-L, TAW: Employed by Genentech at time of study. The other authors declare that no competing interests exist.

### Funding

| Funder | Author |
| --- | --- |
| Genentech | Martin Larhammar<br>Sarah Huntwork-Rodriguez<br>Zhiyu Jiang<br>Hilda Solanoy<br>Arundhati Sengupta Ghosh<br>Bei Wang<br>Joshua S Kaminker<br>Kevin Huang<br>Jeffrey Eastham-Anderson<br>Michael Siu<br>Zora Modrusan<br>Marc Tessier-Lavigne<br>Joseph W Lewcock<br>Trent A Watkins |
| Mission Connect, a program of | Madeline M Farley |

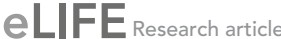

| TIRR Foundation | Trent A Watkins |

The funders had no role in study design, data collection and interpretation, or the decision to submit the work for publication.

## Author contributions

ML, Conceptualization, Data curation, Formal analysis, Investigation, Methodology, Writing—original draft, Project administration, Writing—review and editing; SH-R, Conceptualization, Investigation, experimental design and data acquisition (ISRIB ONC, NGF deprivation model); ZJ, Investigation, experimental design and data acquisition (ONC); HS, ASG, Investigation, experimental design and data acquisition; BW, Investigation, Methodology, experimental design and data acquisition (ONC and IHC); JSK, Data curation, Formal analysis, cross-analysis of microarray data sets; KH, Data curation, Formal analysis, Methodology, RNA-seq data analysis; JE-A, Formal analysis, Methodology, development and execution of quantitative image analysis; MS, Resources, Methodology, synthesis of inhibitors; ZM, Resources, Investigation, Methodology, microarray design and execution; MMF, Investigation, Methodology, method development for harvest of AAV-tranduced regions of retina; MT-L, Conceptualization, Supervision; JWL, Conceptualization, Supervision, Writing—original draft, Project administration, Writing—review and editing; TAW, Conceptualization, Data curation, Formal analysis, Supervision, Funding acquisition, Investigation, Methodology, Writing—original draft, Project administration, Writing—review and editing

## Author ORCIDs

Martin Larhammar, http://orcid.org/0000-0002-1547-6760
Joseph W Lewcock, http://orcid.org/0000-0003-3012-7881
Trent A Watkins, http://orcid.org/0000-0001-6723-3712

## Ethics

Animal experimentation: This study was performed in strict accordance with the recommendations in the Guide for the Care and Use of Laboratory Animals of the National Institutes of Health. All of the animals were handled according to approved institutional animal care and use committee (IACUC) protocols of Genentech, Inc. (Protocols 13-0354B, 13-1675A, TH14-1072, 15-1593, 13-2405, 15-0578, 15-2909, 16-3668, 16-0037, and TH17-0295) or the Baylor College of Medicine (Protocol AN-7208). All surgery was performed under isoflurane anesthesia, and every effort was made to minimize suffering, including the appropriate use of analgesics.

## Additional files

### Major datasets

The following datasets were generated:

| Author(s) | Year | Dataset title | Dataset URL | Database, license, and accessibility information |
|---|---|---|---|---|
| Huang K | 2017 | Response in the NGF withdrawal model | https://www.ncbi.nlm.nih.gov/geo/query/acc.cgi?acc=GSE95672 | Publicly available at the Gene Expression Omnibus (accession no. GSE95672) |
| Huang K | 2017 | Expression changes in L4 DRG initiated by sciatic nerve injury | https://www.ncbi.nlm.nih.gov/geo/query/acc.cgi?acc=GSE96051 | Publicly available at the Gene Expression Omnibus (accession no. GSE96051) |

The following previously published dataset was used:

| Author(s) | Year | Dataset title | Dataset URL | Database, license, and accessibility information |
|---|---|---|---|---|
| Huang K | 2017 | Dlk response to optic nerve injury | https://www.ncbi.nlm. | Publicly available at |

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
