## [Decision Letter]

Thank you for submitting your article "Dual Leucine Zipper Kinase-dependent PERK activation contributes to neuronal degeneration following insult" for consideration by *eLife*. Your article has been reviewed by one peer reviewer, and the evaluation has been overseen by a Reviewing Editor and Marianne Bronner as the Senior Editor. The reviewers have opted to remain anonymous.

The reviewers found the body of work interesting and well performed, however, the transitions from one model to another was not clear. More importantly, the data in the current manuscript supports a role for DLK, PERK and ATF4 in ISR. However, the authors do not fully demonstrate this signaling cascade in a single ISR model.

Required revisions:

1) The authors have a significant amount of cell culture and in vivo data and combine the data from three different paradigms ONC, SNC and DRG cultures to conclude PERK elevates ATF4 which targets *4ebp1* and *Chac1*. As a consequence, the paper is very hard to follow and it's difficult to compare between different experimental paradigms. They focus on the similarities in mechanism but it would also be useful to comment on differences between the different paradigms.

2) The microarray data (Figure 1 and Figure 1—figure supplement 1) come from a previous publication. It is surprising that the authors haven't repeated this with more current approaches like RNA-seq which might identify additional pathway components. In particular, single cell RNA-seq may circumvent the problems of heterogeneity in their in vivo models, especially if these could be combined with loss of function studies.

3) Data from Figure 2 nicely demonstrates ATF4 deficient mice display a decrease in *4ebp1* and *Chac1* (known targets) in only SNC, yet data from Figure 4 demonstrates PERK null mice decrease *4ebp1* and *Chac1* in only ONC. To fully validate the manuscripts conclusion, the authors should determine if PERK null mice display a decrease in ATF4, *4ebp1* and *Chac1* following SNC and/or determine if ATF4 null mice display a decrease in *4ebp1* and *Chac1* following ONC as was done nicely in Figure 5 for DLK.

4) Finally, can the authors comment/test whether ISRIB can in fact protect RGC survival in the ONC model?

---

## [Author Response]

*Required revisions:*

*1) The authors have a significant amount of cell culture and* in vivo *data and combine the data from three different paradigms ONC, SNC and DRG cultures to conclude PERK elevates ATF4 which targets 4ebp1 and Chac1. As a consequence, the paper is very hard to follow and it's difficult to compare between different experimental paradigms. They focus on the similarities in mechanism but it would also be useful to comment on differences between the different paradigms.*

This comment has been very helpful in the revision of our manuscript. One of the strengths of our study – the broad conclusions possible by the use of multiple models and the opportunity to utilize the most advantageous model for each experimental question (validating across models wherever possible) – also had a downside in reduced clarity of presentation. Moreover, while the similarities across diverse models indicate that PERK activation is a common feature of DLK activation, our focus on that aspect may obscure important differences among these models. We have worked to address both of these concerns in the updated manuscript by making the following changes:

1) In the revised manuscript, we have made a number of edits to the text in order to more clearly describe these results and improve both the transitions and comparisons of the results from each model.

2) In the original manuscript, we organized figures largely by experimental question rather than by model. This organization was driven in part by the cross-model (SNC vs. ONC) expression analysis that was the initial hypothesis generator and in part by the technical limitations of demonstrating the entire pathway in a single model. With our new data addressing this last difficulty (see point #3 below), the ONC model takes an even more prominent role in the revised manuscript. We have therefore made some adjustments to the ordering of the figure panels to group the ONC experiments demonstrating the pathway (Figure 7) and its consequences more closely together (Figure 8).

3) We have highlighted some differences observed between models. For example, we have included new qPCR data, which indicate that ISR disruption impacts not only *Chac1* and *4ebp1* upregulation in the NGF withdrawal model but also *Chop* and *Ppp1r15a* upregulation (the latter two being unaffected in the nerve crush models). This suggests a difference in the relative contributions of PERK and *JNK* signaling to their regulation of some mRNAs in each model (Figure 2).

4) In describing our new RNA-seq data in the NGF deprivation model (see point #2 below for more detail), we compare the differences in global expression changes in this model as compared to SNC and ONC. These distinctions help to emphasize the important differences in cell type, stage of development, and type of insult among these models, perhaps making the general commonality of ISR activation even more striking. (Figure 5—figure supplement 1 and Figure 5—figure supplement 2).

*2) The microarray data (Figure 1 and Figure 1—figure supplement 1) come from a previous publication. It is surprising that the authors haven't repeated this with more current approaches like RNA-seq which might identify additional pathway components. In particular, single cell RNA-seq may circumvent the problems of heterogeneity in their in vivo models, especially if these could be combined with loss of function studies.*

We agree that RNA-seq can serve as a valuable tool. Indeed, several high quality expression SNC and ONC profiling studies, including RNA-seq, have been reported in recent years (Hu et al., 2016; Li et al., 2015; Michaelevski et al., 2010; Whitworth et al., 2016; Yasuda et al., 2016; Yasuda et al., 2014; Yi et al., 2015). Rather than repeating these, we have carefully considered how RNA-seq could most directly inform our current study. We now present a new RNA-seq experiment that addresses a question that follows naturally from our work: Which subset of DLK-mediated expression changes can be attributed, at least in part, to activation of the ISR?The rationale and results of this experiment are detailed below, and these new data significantly extend our understanding of ISR-dependent expression changes following acute neuronal injury.

Our initial submission included a cross-analysis of our unpublished SNC microarray and our previously published ONC microarray. Although neither is uniquely valuable, this analysis was presented by way of introduction to the hypothesis that the ISR could be engaged in acute neuronal insults. We note that the same hypothesis could have been generated by assessment of various published expression profiles (Yang et al., 2016; Yasuda et al., 2014).

To directly extend our current study, we focused on the NGF deprivation model as a system for which: (1) there is relatively little published expression profiling (Maor-Nof et al., 2016); (2) there is a relatively homogenous neuron-enriched population of cells, and (3) pharmacological manipulation of DLK and the ISR can be performed without assessment of complicated in vivo pharmacodynamics. We utilized this model to determine which aspects of the DLK-mediated stress response are ISR-dependent and which are ISR-independent. Because this question arises directly from our discovery that ISR activation is DLK/JNK-dependent (presented in Figure 3–Figure 4), it is presented in a new Figure 6 rather than in conjunction with the hypothesis-generating ONC and SNC expression profiling presented in Figure 1. This experiment yielded a number of interesting results, including:

1) Further support for the upregulation of ISR-related genes following NGF withdrawal (Figure 2 and Figure 5).

2) Identification of ISR-dependent expression changes as largely DLK-dependent, consistent with our findings across all three models (Figure 5).

3) Novel candidates for DLK- and ISR-dependent expression changes, some of which are shared among all models and some of which appear to be characteristic only of the NGF-deprivation model (Figure 5).

4) Evaluation of the relative proportions of the DLK-mediated response that is ISRIB-sensitive and ISRIB-insensitive, which suggests that both groups include a broad range of expression changes (Figure 5).

The RNA-seq will be provided to the public following publication through GEO but a link has also been created to allow review of record GSE95672 while it remains in private status.

*3) Data from Figure 2 nicely demonstrates ATF4 deficient mice display a decrease in 4ebp1 and Chac1 (known targets) in only SNC, yet data from Figure 4 demonstrates PERK null mice decrease 4ebp1 and Chac1 in only ONC. To fully validate the manuscripts conclusion, the authors should determine if PERK null mice display a decrease in ATF4, 4ebp1 and Chac1 following SNC and/or determine if ATF4 null mice display a decrease in 4ebp1 and Chac1 following ONC as was done nicely in Figure 5 for DLK.*

These suggested reciprocal experiments are certainly appealing, and we regret not fully explaining the technical challenges that precludes their inclusion. We have now completed alternative experiments to fill in the gaps and demonstrate the complete the pathway in a single model.

Specifically, the exclusion of these precise experiments is the result of limitations of the genetic models. ATF4-deficient mice have severe microphthalmia (Masuoka and Townes, 2002), a phenotype that is incompatible with ONC (noted in the subsection “PERK signaling contributes to neurodegeneration after axon injury”). We therefore only utilized these mice for SNC. To overcome this limitation and complete the demonstration of the entire pathway in the ONC model, we have included new experiments demonstrating that ISRIB treatment (Figure 7) and AAV-shRNA-mediated partial knockdown of ATF4 in retina (Figure 7, Figure 7—figure supplement 4) reduces *Chac1* mRNA induction.

Conversely, we utilized the PERK cKO mice only for ONC experiments because of inefficient knockout of PERK in DRG neurons. Although the Synapsin1-Cre transgene efficiently drives recombination in RGCs (Figure 7—figure supplement 2), we found that it is effective only in *large-diameter* DRG neurons (Figure 1—figure supplement 4), limiting the utility of SNC experiments that require whole DRG lysates for analysis (e.g., immunoblots for ATF4, qPCR for ATF4 target genes). With no appropriate Cre lines available on a reasonable timeline, we considered alternative approaches, each with its own limitation: IHC for ATF4 (which lacked specificity, with signal still observed in ATF4-deficient tissue); AAV injection (engages stress responses on its own); or dissociation of DRG for single cell analysis (also engages stress responses on its own). Given these limitations, we have pursued pharmacological approaches. We found that PERK inhibitors reduced injury-induced ATF4 upregulation following SNC (not shown), but poor drug tolerance and moderate induction of ATF4 by PERKi even in the absence of nerve crush make those results unsuitable for confident interpretation. We instead now provide data from ISRIB-treated mice following SNC, which gave results consistent with our data utilizing ATF4-deficient mice and expand the description of the pathway in this model (Figure 2).

Taken together, these new SNC and ONC experiments provide further evidence of a consistent pathway across models.

*4) Finally, can the authors comment/test whether ISRIB can in fact protect RGC survival in the ONC model?*

We share the interest in this question, and we now report the results of a new experiment that demonstrates partial neuroprotection by ISRIB in the ONC model (Figure 8, Figure 8—figure supplement 1). Technical differences, including the complicated in vivo pharmacodynamics of long-term inhibitor dosing, suggest cautious interpretation in direct comparisons with the efficacy of genetic disruption. Nevertheless, these results are consistent with our findings in the PERK cKO mice, providing further support for the involvement of the ISR in the death of RGCs following ONC.